# Embryo-derive TNF promotes decidualization via fibroblast activation

Si-Ting Chen[1,2], Wen-Wen Shi[2], Yu-Qian Lin[2], Zhen-Shan Yang[2], Ying Wang[2], Meng-Yuan Li[2], Yue Li[2], Ai-Xia Liu[3], Yali Hu[4]*, Zeng-Ming Yang[1]*

[1]Key Laboratory of Plateau Mountain Animal Genetics, Breeding and Reproduction, Ministry of Education, Guizhou University; College of Animal Science, Guizhou University, Guiyang, China; [2]College of Veterinary Medicine, South China Agricultural University, Guangzhou, China; [3]Department of Reproductive Endocrinology, Women's Hospital, Zhejiang University School of Medicine, Zhejiang, China; [4]Department of Obstetrics and Gynecology, The Affiliated Drum Tower Hospital of Nanjing University Medical School, Nanjing, China

*For correspondence:
glyyhuyali@163.com (YH);
yangzm@gzu.edu.cn (Z-MY)

**Competing interest:** The authors declare that no competing interests exist.

**Abstract** Decidualization is a process in which endometrial stromal fibroblasts differentiate into specialized secretory decidual cells and essential for the successful establishment of pregnancy. The underlying mechanism during decidualization still remains poorly defined. Because decidualization and fibroblast activation share similar characteristics, this study was to examine whether fibroblast activation is involved in decidualization. In our study, fibroblast activation-related markers are obviously detected in pregnant decidua and under in vitro decidualization. ACTIVIN A secreted under fibroblast activation promotes in vitro decidualization. We showed that arachidonic acid released from uterine luminal epithelium can induce fibroblast activation and decidualization through $PGI_2$ and its nuclear receptor PPARδ. Based on the significant difference of fibroblast activation-related markers between pregnant and pseudopregnant mice, we found that embryo-derived TNF promotes $CPLA_{2\alpha}$ phosphorylation and arachidonic acid release from luminal epithelium. Fibroblast activation is also detected under human in vitro decidualization. Similar arachidonic acid-$PGI_2$-PPARδ-ACTIVIN A pathway is conserved in human endometrium. Collectively, our data indicate that embryo-derived TNF promotes $CPLA_{2\alpha}$ phosphorylation and arachidonic acid release from luminal epithelium to induce fibroblast activation and decidualization.

## Editor's evaluation

The authors provide novel evidence for a connection between fibroblast activation and eutherian stromal decidualization. This important work substantially advances our understanding of decidua biology and its contribution to pregnancy. The authors are using solid evidence to support the findings.

## Introduction

Endometrial stromal fibroblasts undergo decidualization to form specialized secretory decidual cells. Decidualization is characterized by significant proliferation, differentiation, and epithelial transition of endometrial stromal cells (*Gellersen and Brosens, 2014*; *Tan et al., 2004*). In mice, decidualization occurs only after the onset of implantation. However, decidualization can also be induced by artificial stimuli (*McConaha et al., 2011*). In contrast to mice, human decidualization does not require embryo implantation and is driven by the rise of postovulatory progesterone and local cyclic adenosine monophosphate (cAMP) from endometrial stromal cells (*Gellersen and Brosens, 2014*).

Although decidualization is essential to the successful establishment of pregnancy, the underlying mechanism during decidualization is still poorly defined. Accumulating evidence indicates that decidualization and fibroblast activation share similar characteristics. During primate decidualization, αSMA increases at the initial stage and decreases at the final differentiation stage (*Kim et al., 1999*). αSMA is also strongly expressed in rat decidual cells (*Venuto et al., 2008*). αSMA expression in interstitial fibroblasts during pregnancy correlates with the onset of the decidual process (*Fazleabas et al., 1999*). Meanwhile, αSMA is strongly expressed in myofibroblast and often recognized as the marker of fibroblast activation (*Fujigaki et al., 2005*; *Angelini et al., 2020*). ATP and uric acid belong to damage-associated molecular patterns (DAMPs), which are released following tissue injury or cell death (*Chen and Nuñez, 2010*). We recently demonstrated that either ATP or uric acid can induce mouse decidualization (*Gu et al., 2020*; *Zhu et al., 2021*). It has been shown that ATP or uric acid can also stimulate fibroblast activation (*Dolmatova et al., 2012*; *Bao et al., 2018*).

Fibroblasts are the most numerous cells in connective tissue. Fibroblast activation refers to the process in which dormant quiescent fibroblasts in tissues are stimulated to form functionally active fibroblasts and perform different functions. Fibroblasts synthesize the extracellular matrix of connective tissue and play a key role in maintaining the structural integrity of most tissues (*Yoshida, 2020*; *Tracy et al., 2016*; *Enzerink and Vaheri, 2011*). In healthy and intact tissues, fibroblasts remain a dormant and non-proliferating state. Upon stimulation, dormant fibroblasts acquire contractile properties by inducing the formation of stress fibers, resulting in the formation of myofibroblasts (*Pakshir et al., 2020*; *Tomasek et al., 2002*). The myofibroblast is a specialized fibroblast expressing α-smooth muscle actin (α-SMA; *Angelini et al., 2020*). α-SMA is also a marker of cancer-associated fibroblasts (CAFs; *Nurmik et al., 2020*). In tumor tissues, activated stromal fibroblasts are called as CAFs and show similar characteristics with myofibroblasts (*Shimura, 2021*). Under the stimulation of cytokines and growth factors, fibroblasts will transform into myofibroblasts at the initial stage and further differentiate to into functionally fibroblasts. Activated fibroblasts are characterized by expressing α-SMA, FAP, Vimentin, Desmin, S100A4(also called FSP1), Tenascin C (TNC), periostin, and SPARC (*Kuzet and Gaggioli, 2016*).

In adult tissues, the human endometrium undergoes cyclical shedding and bleeding, scar-free repair and regeneration in subsequent cycles (*Martin, 2007*; *Salamonsen et al., 2021*). In the uterus, the mesenchyme accounts for about 30–35% of the main uterine cell types. Six cell types have been identified in human endometrium, including stromal fibroblasts, endothelial cells, macrophages, uNK, lymphocytes, epithelial cells, and smooth muscle cells (*Wang et al., 2020*). Based on a recent single-cell transcriptomic analysis of human endometrium, stromal cells were the most abundant cell type in the endometrium (*Lv et al., 2022b*). As mouse blastocyst begins to adhere to the luminal epithelium on day 4 of pregnancy, the activated embryo secretes a series of factors to remodel the stationary fibroblasts through epithelial cells (*Li et al., 2015*). Whether fibroblast activation is involved in embryo implantation and decidualization is still unknown.

Arachidonic acid (AA) is the biosynthetic precursor of prostaglandins in the cell membrane. CPLA$_{2\alpha}$ is encoded by the *Pla2g4a* gene and a major provider of AA. *Pla2g4a* knockout mice show uneven embryo distribution and reduced litter size, suggesting that maternal uterine CPLA2α is critical for successful embryo implantation (*Song et al., 2002*). Cyclooxygenase COX is the rate-limiting enzyme for prostaglandin synthesis. COX-1 knockout mice are fertile, with only birth defects (*Lim and Dey, 1997*). COX-2 knockout on C57BL/6 J/129 background mice results in impaired implantation, and decidualization, indicating the important role of COX-2 during embryo implantation and decidualization (*Lim and Dey, 1997*). Prostaglandin I$_2$ (PGI$_2$) and prostaglandin E$_2$ (PGE$_2$) are the two most abundant prostaglandins at the embryo implantation site (*Lim et al., 1999*). PGI$_2$ derived from COX-2 and prostacyclin synthase (PGIS) can regulate embryo implantation through peroxisome proliferators-activated receptor δ (PPARδ; *Lim et al., 1999*). Both embryonic implantation and decidualization are abnormal in *PPARδ-/-* mice (*Wang et al., 2007*). Although prostaglandins are required for embryo implantation and decidualization, whether prostaglandins are involved in fibroblast activation during early pregnancy remains unclear.

This study was to analyze whether fibroblast activation is involved in mouse and human decidualization, and why fibrosis does not occur during normal early pregnancy. In this study, the multiple markers of fibroblast activation were detected during mouse in vivo and in vitro decidualization. Embryo-derived TNF was able to induce the phosphorylation of CPLA$_{2\alpha}$ in luminal epithelium, which

liberated AA into uterine stroma to promote fibroblast activation through COX2-PGI$_2$-PPARδ pathway for inducing decidualization. The underlying mechanism of fibroblast activation was also conserved during human in vitro decidualization. This study offers novel insights into the function of fibroblasts during embryo implantation and decidualization.

## Results

### Fibroblast activation is present during mouse early pregnancy

S100A4, αSMA, TNC, and SPARC are well-recognized markers of fibroblast activation (*Kuzet and Gaggioli, 2016*; *Prime et al., 2017*). Immunofluorescence and immunohistochemistry showed that TNC, SPARC, and S100A4 were mainly localized in the primary decidual region after embryo implantation. SPARC immunofluorescence was strongly observed in subluminal stromal cells from D4.5 to D5.5. However, TNC immunostaining was strongly detected in subluminal stromal cells on D4, D4.5, and D5, but disappeared on D5.5. S100A4 immunostaining wasn't detected in mouse uterus on D4 and D4.5 of pregnancy, but detected in subluminal stromal cells at implantation sites on D5 and D5.5 (*Figure 1A*). Under in vitro decidualization, the protein levels of αSMA, TNC and SPARC were significantly increased by 4.41, 3.25, and 2.69-folds compared with the control group, respectively (*Figure 1B*). These results suggested that fibroblast activation should be present under in vivo and in vitro decidualization.

### Fibroblast activation promotes decidualization by secreting ACTIVIN A

BMP2 and WNT4 are essential to mouse decidualization (*Lee et al., 2007*; *Li et al., 2007*). E2F8 and CYCLIN D3 are markers of polyploidy during mouse decidualization (*Qi et al., 2015*). It has been shown that activated fibroblasts can secrete S100A4, SPARC, TNC, and ACTIVIN A (*Kuzet and Gaggioli, 2016*; *Truffi et al., 2020*). In order to examine whether fibroblast activation is involved in mouse decidualization, mouse stromal cells were treated with S100A4, SPARC, TNC, and ACTIVIN A to induce in vitro decidualization, respectively. TNC treatment upregulated the protein levels of WNT4 and CYCLIN D3, but had no obvious effects on BMP2, E2F8, and *Prl8a2* (*Figure 2A and B*). S100A4 treatment upregulated BMP2 and WNT4 protein levels by 1.76- and 1.42-folds, respectively, but had no obvious effects on E2F8, CYCLIN D3, and *Prl8a2* (*Figure 2C and D*). SPARC treatment on mouse stromal cells had no significant effect on decidualization markers (*Figure 2E and F*). Because ACTIVIN A is secreted under fibroblast activation (*Loomans and Andl, 2014*), we examined ACTIVIN A protein levels in the uteri on D4 and D4.5 of pregnancy and pseudopregnant (PD4, PD4.5) mice, respectively. The level of ACTIVIN A protein on D4 was 1.75-folds that of PD4 mice, and the level of ACTIVIN A protein on D4.5 was 3.77-folds that of PD4.5 mice. The protein level of ACTIVIN A on D4.5 was 1.88-folds higher than that on D4 (*Figure 2G*), indicating that the secretion of ACTIVIN A increased after embryo implantation. After mouse stromal cells were treated with ACTIVIN A, all the protein levels of BMP2, WNT4, E2F8, CYCLIN D3, and *Prl8a2* were obviously up-regulated 1.35, 2.36, 3.11, 1.90, and 6.29 folds, respectively (*Figure 2H, I*). Under in vitro decidualization, ACTIVIN A treatment also increased all the protein levels of BMP2, WNT4, E2F8, and CYCLIN D3 by 1.52, 1.43, 3.89, and 1.36 folds, respectively (*Figure 2J*). These results suggest a positive correlation between ACTIVIN A and decidualization.

### PGI$_2$ promotes fibroblast activation and decidualization through PPARδ pathway

Although we just showed the presence of fibroblast activation during decidualization, what initiates fibroblast activation is still unknown. A previous study indicated that PGE$_2$ and PGI$_2$ are the most abundant prostaglandins at implantation sites in mouse uterus (*Lim et al., 1999*). Both PGE$_2$ and PGI$_2$ are essential to mouse decidualization (*Lim et al., 1999*; *Holmes and Gordashko, 1980*). When mouse stromal cells were treated with PGE$_2$ in the absence or presence of progesterone (P4), PGE$_2$ had no obvious effects on the protein levels of αSMA, this was despite significant increases in TNC and SPARC (*Figure 3A*). However, ILOPROST, a PGI$_2$ analog (*Huang et al., 2002*), up-regulated the protein levels of TNC, αSMA and SPARC (2.65, 1.80, and 1.53 folds). PGI$_2$ membrane receptor (IP) (*Namba et al., 1994*) was down-regulated by ILOPROST treatment, but peroxisome proliferator-activated receptor δ (PPARδ), the PGI$_2$ nuclear receptor, was significantly increased by ILOPROST (2.54 folds) (*Figure 3B*).

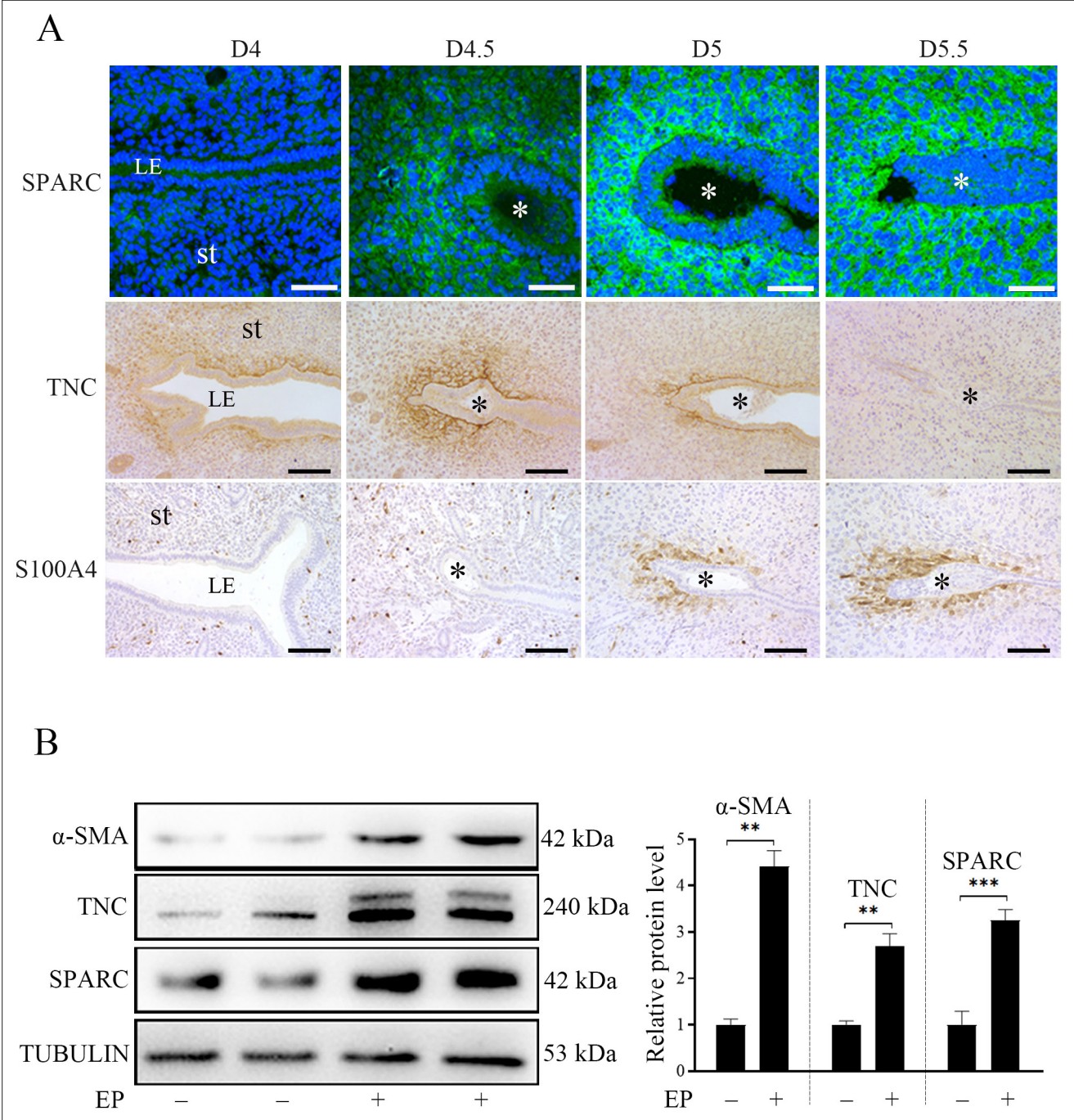

**Figure 1.** The protein localization and levels of markers of fibroblast activation in mouse uteri during early pregnancy. (**A**) Immunofluorescence of SPARC and immunohistochemistry of TNC and S100A4 in mouse uteri on D4 (n=5), D4.5 (n=5), D5 (n=5), and D5.5 (n=5) of pregnancy. LE, luminal epithelium; St, stroma; * Embryo. Scale bar, 50 μm. (**B**) Western blot analysis of α-SMA, SPARC, TNC protein level under in vitro decidualization (EP) for 24 hr. *, p<0.05; **, p<0.01; ***, p<0.001.

The online version of this article includes the following source data for figure 1:

**Source data 1.** Raw data of all western blots from *Figure 1*.

**Source data 2.** Complete and uncropped membranes of all western blots from *Figure 1*.

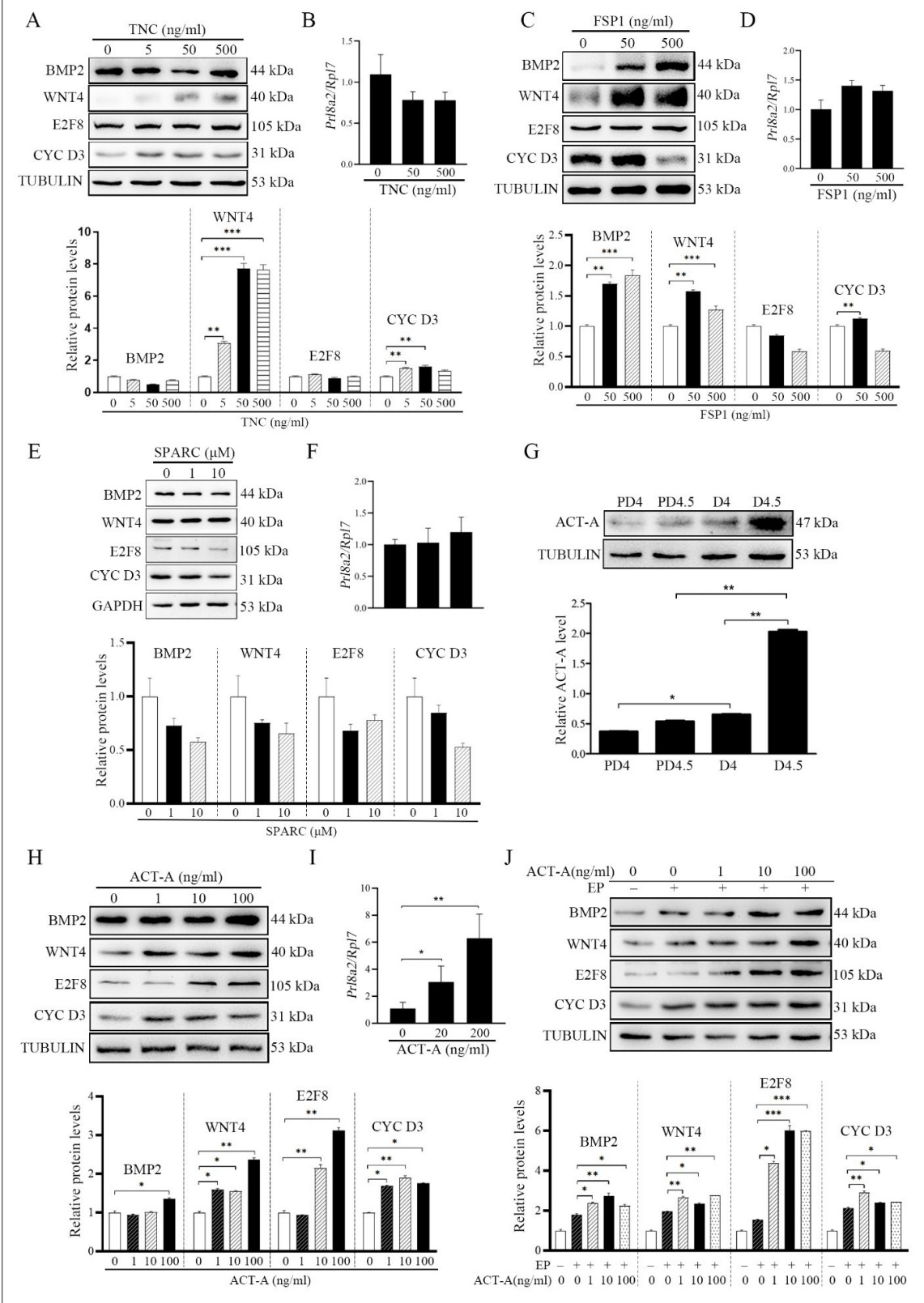

**Figure 2.** Fibroblast activation promotes decidualization by secreting ACTIVIN A. (**A**) Western blot analysis on the effects of TNC on decidualization markers (BMP2, WNT4, E2F8 and CYCLIN D3) after stromal cells were treatment with TNC for 72 hr. (**B**) QPCR analysis of *Prl8a2* mRNA level after mouse stromal cells were treated with TNC for 72 hr. (**C**) Western blot analysisof the effects of S100A4 on decidualization markers after stromal cells were treated with S100A4 for 72 hr. (**D**) QPCR analysis of *Prl8a2* mRNA level after mouse stromal cells were treated with S100A4 for 72 hr. (**E**) Western

*Figure 2 continued on next page*

*Figure 2 continued*

blot analysis on the effects after stromal cells were treated with SPARC for 72 hr. (**F**) QPCR analysis of *Prl8a2* mRNA level after mouse stromal cells were treated SPARC for 72 hr. (**G**) Western blot analysis on ACTIVIN A protein levels in mouse uteri on D4, D4.5, PD4, and PD4.5, respectively. (**H**) Western blot analysis on the effects of ACTIVIN A on decidualization markers after stromal cells were treated with ACTIVIN A for 72 hr. (**I**) QPCR analysis of *Prl8a2* mRNA level after mouse stromal cells were treated with ACTIVIN A for 72 hr. (**J**) Western blot analysis on the effects of ACTIVIN A on decidualization markers after stromal cells were treated with ACTIVIN A for 48 hr under in vitro decidualization. EP, 17β-estradiol+progesterone. All data were is presented as means ± SD. *, p<0.05; **, p<0.01; ***, p<0.001. CYC D3: CYCLIN D3; ACT-A: ACTIVIN A.

The online version of this article includes the following source data for figure 2:

**Source data 1.** Raw data of all western blots from *Figure 2*.

**Source data 2.** Complete and uncropped membranes of all western blots from *Figure 2*.

PPARδ agonist GW501516 was also able to upregulate the protein levels of TNC, αSMA and SPARC by 7.77, 2.81, and 2.03 folds, respectively (*Figure 3C*). Further analysis showed that treatment of stromal cells with 0.1 μM of IP agonist SELEXIPAG (*Asaki et al., 2015*) increased a-SMA and SPARC levels by 1.39 and 2.26 folds, respectively, but had no obvious effect on TNC level. When stromal cells were treated with 1 or 10 μM of SELEXIPAG, the protein levels of a-SMA and SPARC decreased (*Figure 3D*). After stromal cells were treated with either ILOPROST or GW501516, decidualization markers (BMP2, WNT4, E2F8, and CYCLIN D3) were also significantly up-regulated by 1.96, 1.81, 1.33, and 1.87, or 1.73, 5.34, 3.73, and 1.91 folds, respectively (*Figure 3E and G*). The protein levels of WNT4 and CYCLIN D3 were decreased by 10 μM ILOPROST (*Figure 3G*). The mRNA levels of Prl8a2 were also increased by either ILOPROST or GW501516 (*Figure 3F and H*). Compared with control, the protein level of ACTIVIN A was obviously upregulated by 0.1 μM ILOPROST (1.48 folds) (*Figure 3I*). These results showed that PGI$_2$ initiated fibroblast activation and promoted decidualization through the PPARδ pathway.

## Arachidonic acid induces the fibroblasts activation and promotes decidualization through stimulating Activin A secretion

AA is the precursor of prostaglandin biosynthesis and can be liberated from membrane lipids through the phosphorylation of CPLA$_{2α}$ (*Simmons et al., 2004*). CPLA$_{2α}$ is significantly expressed in the luminal epithelium in mouse uterus during peri-implantation period and essential for mouse embryo implantation (*Song et al., 2002*), suggesting that AA should be released from luminal epithelium. After stromal cells were treated with AA, the markers of fibroblast activation (TNC, αSMA, and SPARC) were significantly up-regulated (1.63, 16.00. and 31.20 folds) (*Figure 4A*). Treatment with AA also upregulated the protein levels of decidualization markers (BMP2 by 1.25-folds, WNT4 by 1.27-folds, E2F8 by 15.78-folds and CYCLIN D3 by 6.31-folds) (*Figure 4B*) and the mRNA level of *Prl8a2* (another marker for mouse decidualization) (*Figure 4D*). Under in vitro decidualization, AA treatment also significantly increased the decidualization markers (BMP2 by 3.54-folds, WNT4 by 1.81-folds, E2F8 by 3.09-folds and CYCLIN D3 by 2.68-folds; *Figure 4C*). These results indicated that AA might promote decidualization through fibroblast activation. Additionally, treatment with AA distinctly stimulated the protein levels of COX-2, PGIS, and PPARδ (10.94, 1.99, and 53.03 folds), but had no obvious effects on PGES and IP (*Figure 4E*). The stimulation of AA on PGIS and PPARδ was abrogated by COX-2 inhibitor NS398 (*Figure 4F*). The induction of AA on *Prl8a2* mRNA levels was also suppressed by either COX-2 inhibitor NS398 or PPARδ antagonist GSK0660 (*Figure 4H and I*). AA also significantly induced the protein level of ACTIVIN A and the mRNA level of *Inhba* (encoded for ACTIVIN A; *Figure 4G and J*). Furthermore, the induction of AA on *Prl8a2* expression was abrogated by either ACTIVIN A inhibitor (SB431542) or *Inhba* siRNA (*Figure 4K to M*). These results suggested that AA induced fibroblast activation and promoted decidualization through ACTIVIN A.

## Blastocyst-derived TNF promotes CPLA$_{2α}$ phosphorylation and AA secretion

We just showed that AA could initiate fibroblast activation and induce decidualization. When we examined the concentration of AA in the uterine luminal fluid, the luminal concentration of AA at D4.5 was significantly higher than that on D3 and D4, indicating AA secretion from the uterus just after embryo implantation (*Figure 5A*).

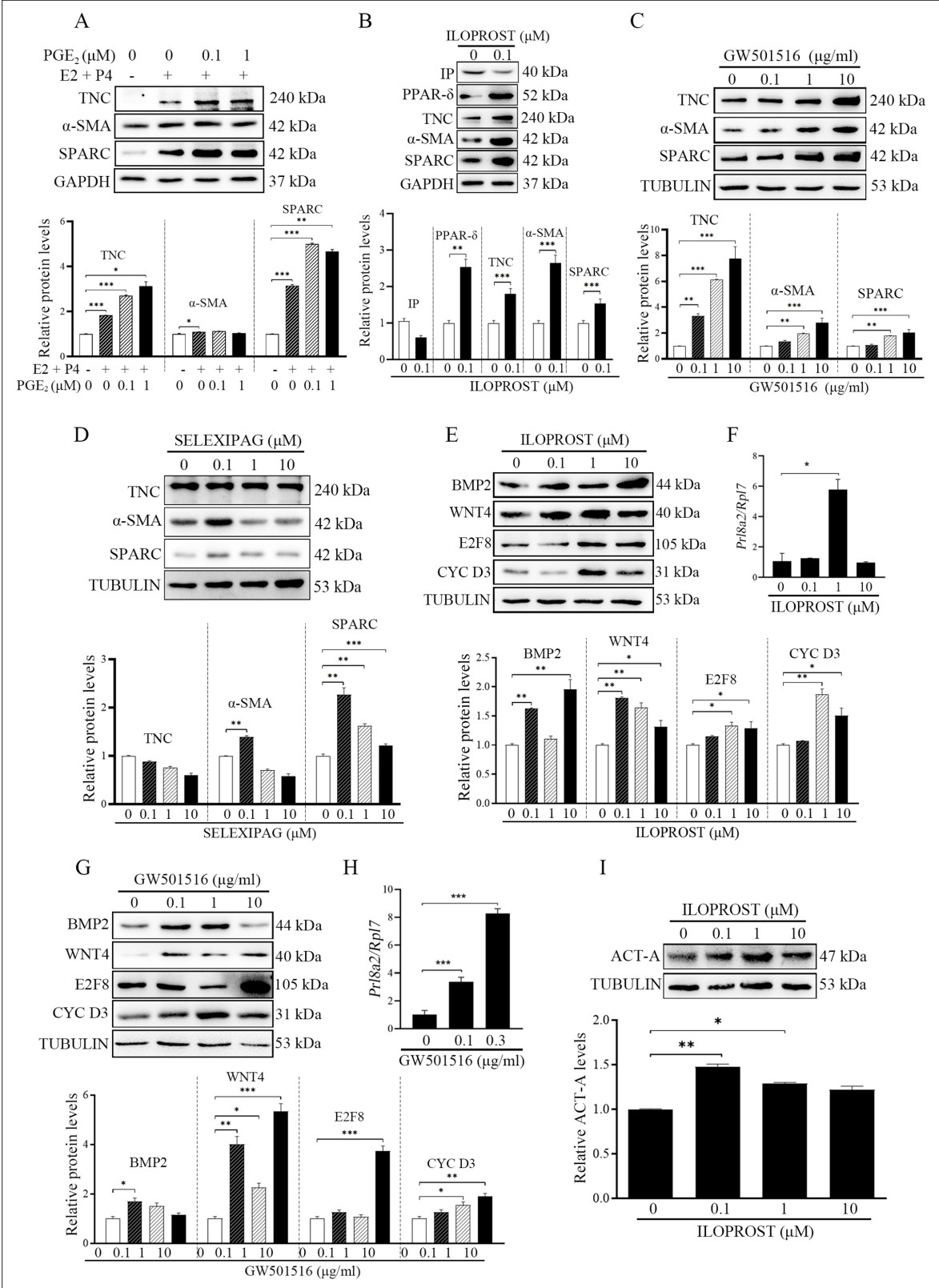

**Figure 3.** Western blot analysis on effects of prostaglandins on fibroblast activation and decidualization in mouse stromal cells. (**A**) The effects of PGE$_2$ on markers of fibroblast activation. (**B**) The effects of ILOPROST, PGI$_2$ analog, on markers of fibroblast activation after stromal cells was treated with ILOPROST for 12 hr. (**C**) The effects of GW501516, PPARδ agonist, on markers of fibroblast activation. (**D**) The effects of SELEXIPA, IP analog, on markers of fibroblast activation. (**E**) The effects of ILOPROST on decidualization markers. (**F**) QPCR analysis of *Prl8a2* mRNA level after mouse stromal cells were

*Figure 3 continued on next page*

*Figure 3 continued*

treated with ILOPROST for 48 hr. (**G**) The effects of GW501516 on decidualization markers. (**H**) QPCR analysis of *Prl8a2* mRNA level after mouse stromal cells were treated with GW501516 for 48 h. (**I**) The effects of ILOPROST on ACTIVIN A protein levels after stromal cells were treated with ILOPROST for 24 hr. CYC D3: CYCLIN D3; ACT-A: ACTIVIN A. *, $p<0.05$; **, $p<0.01$; ***, $p<0.001$.

The online version of this article includes the following source data for figure 3:

**Source data 1.** Raw data of all western blots from *Figure 3*.

**Source data 2.** Complete and uncropped membranes of all western blots from *Figure 3*.

Immunofluorescence also showed that the P-CPLA$_{2\alpha}$ level in luminal epithelium in D4.5 was obviously stronger than that on D4 (*Figure 5B*). Compare with inter-implantation sites, P-CPLA$_{2\alpha}$ immunofluorescence at implantation site on D5 was also stronger (*Figure 5C*). Compared with delayed implantation, P-CPLA$_{2\alpha}$ immunofluorescence in the luminal epithelium was stronger 12 hr after estrogen activation (*Figure 5D*). Western blot also confirmed that P-CPLA$_{2\alpha}$ levels on D4 and D4.5 were significantly higher than PD4 and PD4.5 (*Figure 5E*). Compared with delayed implantation, P-CPLA$_{2\alpha}$ protein levels were also higher 12 hr after estrogen activation (*Figure 5F*). These results suggested that embryos should be involved in CPLA$_{2\alpha}$ phosphorylation.

Furthermore, we examined the markers of fibroblast activation. Compared with delayed implantation, the immunostaining levels of both TNC and S100A4 at implantation sites after estrogen activation were stronger (*Figure 5G*). Western blot also indicated that the protein levels of the markers of fibroblast activation (αSMA by 1.32-folds, TNC by 1.51-folds, and SPARC by 4.0-folds) on D4 were stronger than that on PD4 (*Figure 5H*). This results strongly suggest that embryos were tightly involved in fibroblast activation.

S100A9, HB-EGF and TNF are previously shown to be secreted by blastocysts (*He et al., 2019*; *Haller et al., 2019*; *Hamatani et al., 2004*). When S100A9-soaked blue beads were transferred into PD4 uterine lumen, S100A4 had no obvious change, but TNC immunostaining was increased slightly (*Figure 6A*). When HB-EGF-soaked beads were transferred, S100A4 immunostaining was slightly increased, but TNC immunostaining was increased obviously (*Figure 6B*). However, TNF-soaked beads obviously stimulated the immunostaining levels of both S100A4 and TNC (*Figure 6C*).

After TNF-soaked beads were transferred into PD4 uterine lumen, P-CPLA$_{2\alpha}$ immunofluorescence at luminal epithelium was obviously increased (*Figure 6D*). When the epithelial cells isolated PD4 uterus were treated with TNF, western blot showed that the protein level of P-CPLA$_{2\alpha}$ was significantly upregulated 2.31-folds (*Figure 6E*). To confirm the results from in vitro cultured epithelial cells, mouse uterine epithelial organoids was treated with TNF. We found that the expression of P-CPLA$_{2\alpha}$ was up-regulated 2.49-folds by TNF treatment in mouse uterine epithelial organoids (*Figure 6F*). AA concentration in the cultured medium of epithelial orgoids was also up-regulated 1.88-folds by TNF treatment (*Figure 6G*). These results indicated that embryo-derived TNF was able to promote the phosphorylation of CPLA$_{2\alpha}$ in luminal epithelium for liberating AA into uterine stroma.

## Effects of fibroblast activation on TNF-AA-PGI$_2$ pathway under human in vitro decidualization

After we showed that fibroblast activation was involved in mouse decidualization, we wondered whether fibroblast activation participated in human decidualization. Compared with control group, the protein levels of α-SMA, TNC and SPARC were significantly increased under human in vitro decidualization (1.30, 5.12, and 2.29 folds; *Figure 7A*). Previous studied also indicated that human blastocysts could synthesize and secret TNF (*He et al., 2019*; *Lv et al., 2022a*). When human uterine epithelial Ishikawa cells were co-cultured with human 4003 stromal cells, TNF treatment significantly increased the protein level of P-CPLA$_{2\alpha}$ by 1.59-folds in epithelial Ishikawa cells (*Figure 7B*), and the protein levels of TNC, αSMA, SPARC in 4003 stromal cells (2.71, 2.44, and 1.96 folds) (*Figure 7C*). When human stromal cells were treated with AA, the protein levels of TNC, αSMA, and SPARC were obviously stimulated (*Figure 7D*). Meanwhile, treatment with AA also significantly upregulated the protein levels of COX-2, PGIS, and PPARδ (1.28, 4.69, and 1.60 folds), but had no obvious effects on PGES and IP protein level (*Figure 7E*). After 4003 stromal cells were treated with either PGI$_2$ analogs ILOPROST or PPARδ agonists GW501516, the protein levels of TNC, αSMA, and SPARC were significantly up-regulated compared with control group (2.18, 1.57, and 1.38 or 1.79, 2.86, and 3.73 folds;

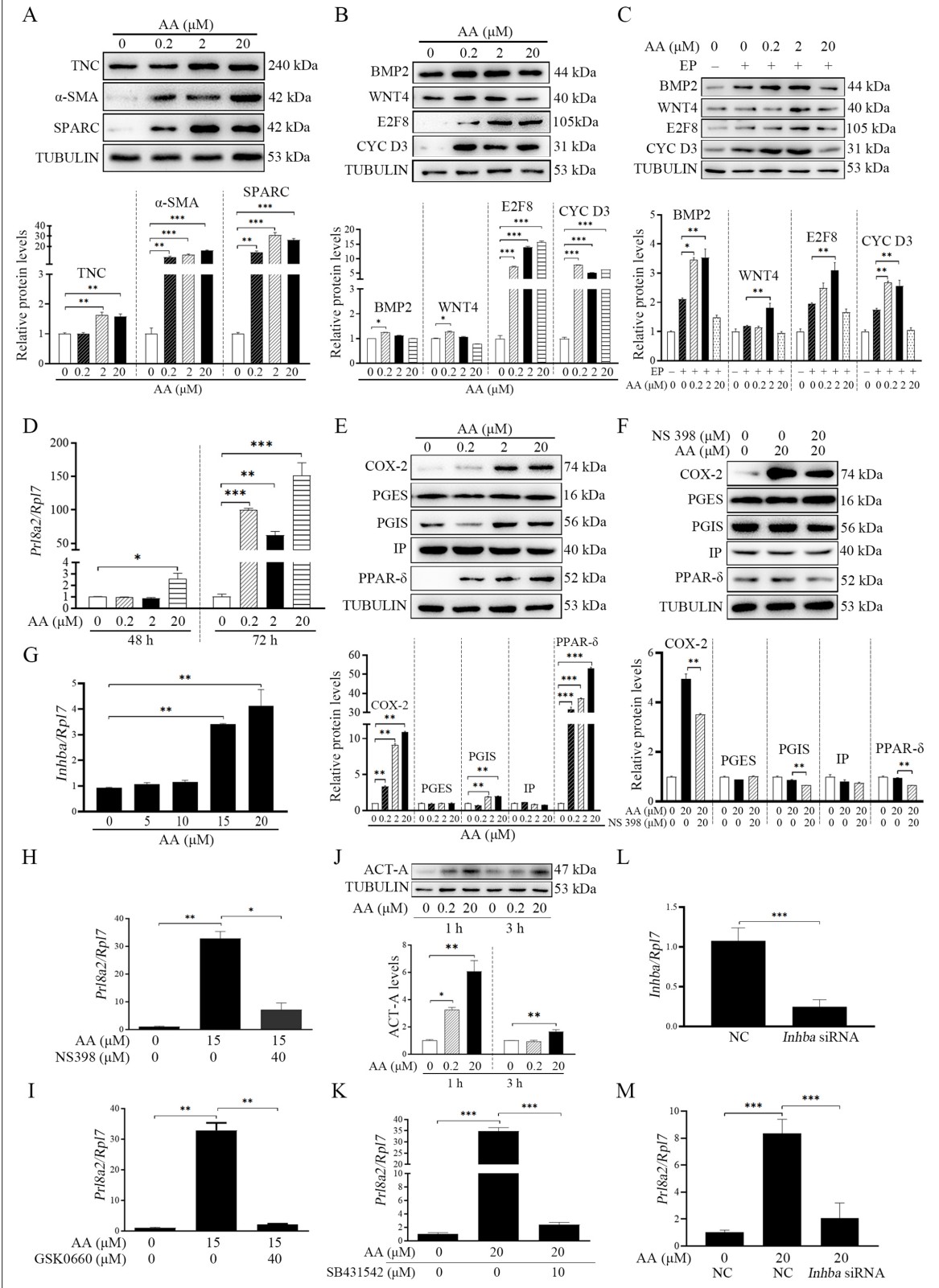

**Figure 4.** Effects of AA on fibroblast activation and decidualization through PGI-PPARδ-ACTIVIN A pathway. (**A**) Western blot analysis on effects of AA on markers of fibroblast activation after stromal cells were treated with AA for 6 hr. (**B**) Western blot analysis on effects of AA on decidualization markers after stromal cells were treated with AA for 48 hr. (**C**) Western blot analysis on effects of AA on decidualization markers after stromal cells were treated with AA for 48 hr under in vitro decidualization. EP, 17β-estradiol+progesterone. (**D**) QPCR analysis of *Prl8a2* mRNA level after stromal cells were treated

*Figure 4 continued on next page*

*Figure 4 continued*

with AA for 72 hr. (**E**) Western blot analysis on effects of AA on COX2, PGES, PGIS, IP and PPARδ protein levels after stromal cells were treated with AA for 6 hr. (**F**) Western blot analysis on effects of NS398 (COX-2 inhibitor) on AA induction of COX2, PGES, PGIS, IP and PPARδ protein levels after stromal cells were treated with AA for 48 hr in the absence or presence of NS398. (**G**) QPCR analysis of *Inhba* mRNA level after stromal cells were treated with AA for 72 hr. (**H**) QPCR analysis of on effects of NS398 on AA induction of *Prl8a2* mRNA level after stromal cells were treated with AA for 72 hr. (**I**) QPCR analysis of on effects of GSK0660 on AA induction of *Prl8a2* mRNA level after stromal cells were treated with AA for 72 hr. (**J**) Western blot analysis on effects of AA on ACTIVIN A protein level after stromal cells were treated with AA for 24 hr. (**K**) QPCR analysis on effects of SB431542 (ACTIVIN A inhibitor) on AA induction of *Prl8a2* mRNA levels. (**L**) QPCR analysis on the interference efficiency of *Inhba*. (**M**) QPCR analysis on effects of *Inhba* siRNAs on AA induction on *Inhba* mRNA level after stromal cells were treated with AA for 72 hr. Data were presented as means ± SD from at least three biological replicates. *, p<0.05; **, p<0.01; ***, p<0.001. CYC D3: CYCLIN D3; ACT-A: ACTIVIN A.

The online version of this article includes the following source data for figure 4:

**Source data 1.** Raw data of all western blots from *Figure 4*.

**Source data 2.** Complete and uncropped membranes of all western blots from *Figure 4*.

*Figure 7F and G*). These results suggested that AA should promote fibroblast activation through PGI$_2$–PPARδ pathway during human decidualization. Treatment with AA also stimulated the mRNA expression of *INHBA* (3.4-folds; ncoded for human ACTIVIN A) (*Figure 7H*), which was significantly abrogated by COX-2 inhibitor NS398 (1.88-folds; *Figure 7I*). Overall, these results indicated that fibroblast activation was also involved in human decidualization in a similar mechanism as in mice.

## AA-PGI$_2$ pathway contributes to human decidualization

Insulin growth factor binding protein 1(*IGFBP1*) and prolactin (*PRL*) are recognized markers for human in vitro decidualization (*Saleh et al., 2011*). When human stromal cells were treated with AA, both *IGFBP1* and *PRL* were significantly increased by 4.80 and 3.87-folds, respectively (*Figure 8A*). Under human in vitro decidualization (MPA +cAMP 100 μM), either ILOPROST or GW501516 could significantly promote the mRNA expression of *IGFBP1* and *PRL* (212.20 and 42.30 folds, or 4363.93 and 36.39 folds) (*Figure 8B and C*). The induction of AA on *IGFBP1* and *PRL* mRNA expression was significantly abrogated by either COX-2 inhibitor NS398 or PPARδ antagonist GSK0660 (1.85 and 4.63 folds; *Figure 8D*). Under human in vitro decidualization (MPA +cAMP 500 μM), the mRNA levels of *IGFBP1* and *PRL* were also significantly up-regulated by ACTIVIN A treatment (6.62 and 1.61 folds; *Figure 8E*). AA-induced increase of *IGFBP1* and *PRL* expression was abrogated by *INHBA* knockdown (*Figure 8F and G*). Taken together, these results suggested that epithelium-derived AA could promote fibroblast activation through COX-2-PGI$_2$-PPARδ pathway and induce decidualization via ACTIVIN A during human decidualization, which was also consistent with the results in mice.

## Discussion

Our study shows that embryo-derived TNF promotes the phosphorylation of CPLA$_{2\alpha}$ and the release of AA from luminal epithelium to induce fibroblast activation and decidualization through ACTIVIN A. We also showed that the pathway underlying fibroblast activation was conserved in mice and humans.

Fibroblast activation could be identified by many markers, including α-SMA, TNC, POSTN, NG-2, PDGF receptor-a/b, S100A4and FAP. These fibroblast markers expressed alone or in combination and could be used to identify distinct subpopulations following fibroblast activation (*Prime et al., 2017*; *Grinnell, 1994*; *Oliver et al., 1999*). Fibroblast activation plays an important role under many physiological or pathological conditions, such as cancer, injury repair and fibrosis. However, the regulation and function of fibroblast activation during decidualization remain unknown. In our study, the regulation and function of fibroblast activation was confirmed by multiple approaches during mouse and human decidualization. A study from single-cell analysis of human decidual cells also suggests the presence of fibroblast activation during human in vitro decidualization (*Stadtmauer and Wagner, 2022*). Based on fibroblast activation markers, we confirmed the existence of fibroblast activation during early pregnancy. Fibroblasts are generally in a dormant and quiescent state in tissues, and are only activated when stimulated (*Pakshir et al., 2020*). In a previous study, mPGES1-derived PGE$_2$ supports the early inflammatory phase of wound healing and may stimulate subsequent fibroblast activation early after damage (*Stratton and Shiwen, 2010*). Although PGE$_2$ has been shown to be essential for mouse decidualization (*Holmes and Gordashko, 1980*; *Baskar et al., 1981*), PGE$_2$ was ineffective in

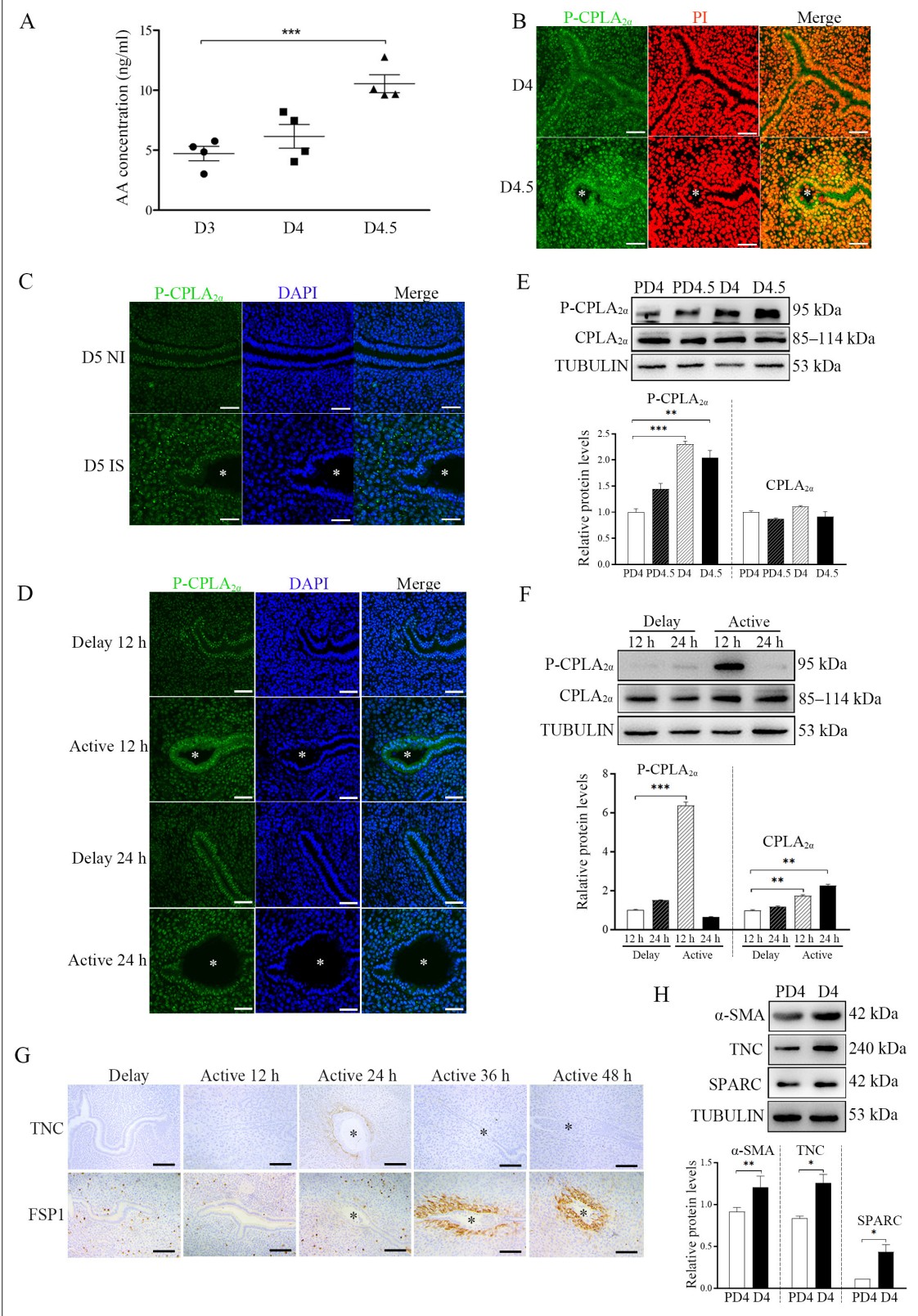

**Figure 5.** The involvement of blastocysts in fibroblast activation during early pregnancy. (**A**) AA concentration in uterine luminal fluid flushed on D3 (n=20 mice), D4 (n=20 mice), and D4.5 (n=20 mice) of pregnancy. (**B**) P-CPLA2α immunofluorescence in mouse uteri on D4 (n=6) and D4.5 (n=6). * Embryo. Scale bar = 50 µm. (**C**) P-CPLA2α immunofluorescence in mouse uteri at implantation sites and inter-implantation sites on D5 (n=6 mice). * Embryo. NI, inter-implantation site; IS, implantation site. Scale bar = 50 µm. (**D**) p-CPLA2α immunofluorescence of in mouse uteri 12 and 24 h after

*Figure 5 continued on next page*

Figure 5 continued

delayed implantation was activated by estrogen treatment, respectively (n=4 mice). * Embryo. Scale bar = 0 μm. (**E**) Western blot analysis of CPLA$_{2\alpha}$ and P-CPLA$_{2\alpha}$ protein levels in mouse uteri on D4, D4.5 PD4 and PD4.5 (n=4 mice), respectively. (**F**) Western blot analysis of CPLA$_{2\alpha}$ and P-CPLA$_{2\alpha}$ protein levels in mouse uteri 12 and 24 hr after delayed implantation was activated by estrogen treatment (n=4 mice). (**G**) Immunostaining of TNC and S100A4 in mouse uteri 12, 24, 36, and 48 hr after delayed implantation was activated by estrogen treatment (n=4 mice). * Embryo. (**H**) Western blot analysis α-SMA, TNC, and SPARC protein levels in mouse uteri on D4 and PD4 (n=4 mice). *, p<0.05; **, p<0.01; ***, p<0.001.

The online version of this article includes the following source data for figure 5:

**Source data 1.** Raw data of all western blots from *Figure 5*.

**Source data 2.** Complete and uncropped membranes of all western blots from *Figure 5*.

activating fibroblast activation in our study. Additionally, PGE2 has no obvious effect on Prl8a2 mRNA level under mouse in vitro decidualization. Our study indicated that AA from luminal epithelium was able to induce fibroblast activation via PGI$_2$-PPARδ pathway. Previous studies indicated that AA could promote mouse decidualization (*Tessier-Prigent et al., 1999*; *Handwerger et al., 1981*). PGI$_2$ has been identified to be essential for mouse decidualization via PPARδ (*Lim et al., 1999*). It is shown that multiple factors are able to induce in vitro decidualization (*Gellersen and Brosens, 2014*). In our study, AA-PGI$_2$ pathway is involved in stimulating decidualization. previous studies indicate that PGI$_2$ can also activate a cAMP signaling pathway (*Beck et al., 2014*; *Fetalvero et al., 2008*). It is possible that AA may also act on decidualization through cAMP. Our study also confirmed AA secretion during either mouse or human in vitro decidualization, which may have positive feedback during in vitro decidualization. In our study, AA concentration in uterine lumen was significantly increased during embryo attachment. AA, PGI$_2$ or PPARδ agonist was able to induce fibroblast activation.

Activated fibroblasts play key roles in the injury response, tumorigenesis, fibrosis, and inflammation through secreting different factors in different physiological or pathological processes (*Avery et al., 2018*). Myofibroblast marker α-SMA has been shown to be one of the important markers of early decidualization in primates (*Kim et al., 1999*), and sterile inflammatory secretion of products such as ATP and uric acid after injury has been shown to stimulate fibroblast activation (*Dolmatova et al., 2012*; *Bao et al., 2018*) and uterine decidualization (*Gu et al., 2020*; *Zhu et al., 2021*), so we speculate that fibroblast activation is strongly correlated with decidualization. Because α-SMA, TNC, SPARC, S100A4, and ACTIVIN A were all identified during fibroblast activation, we examined the role of each of these markers during mouse decidualization. We found that only ACTIVIN A was able to induce mouse in vitro decidualization. The function of ACTIVIN A during human decidualization was also confirmed in our study. ACTIVIN A has been shown to be important for human decidualization (*Zhao et al., 2018*). ACTIVIN A, its functional receptors, and binding proteins, are abundant in human endometrium (*He et al., 1999*). Our study indicated that AA-PGI$_2$-PPARδ axis stimulated fibroblast activation and induced decidualization through ACTIVIN A.

The adequate molecular interaction between the endometrium and the blastocyst is critical for successful implantation and decidualization (*Massimiani et al., 2019*; *Latifi et al., 2018*). In our study, there was a big difference of both markers of fibroblast activation and P-CPLA2α between pregnancy and pseudopregnancy, and between delayed and activated implantation, strongly suggesting the involvement of embryos in these processes. Although S100A9, HB-EGF, and TNF are previously shown to be secreted by blastocysts (*He et al., 2019*; *Haller et al., 2019*; *Hamatani et al., 2004*), TNF was the only one to stimulate both S100A4 and TNC, and to phosphorylate CPLA$_{2\alpha}$ in our study. CPLA$_{2\alpha}$ is a major provider of AA. *Pla2g4a–/–*mice results in deferred implantation and deranged gestational development (*Song et al., 2002*). We also showed that AA concentration in luminal fluid was significantly increased in day 4 evening when the embryos just implanted. TNF is present in the reactivated blastocyst and human blastocyst (*He et al., 2019*; *Lv et al., 2022a*), and may play a critical role during embryo implantation (*You et al., 2021*). In our study, we confirmed that TNF stimulated the phosphorylation of CPLA$_{2\alpha}$ and AA release from luminal epithelium. A proper interaction between embryo attachment and decidualization is critical for successful pregnancy.

Activated fibroblasts play an important role in many physiological and pathologic processes. Excessive fibroblast activation can lead to fibrosis (*Truffi et al., 2020*; *Giusti et al., 2022*; *Yeo et al., 2018*). Fibroblast activation may be present and balanced during normal pregnancy without ultimately leading to fibrosis or other diseases. S100A4 is hypomethylated and overexpressed in grade 3 endometrioid tumors compared with benign endometrium (*Xie et al., 2007*). Genetic and proteomic analysis of

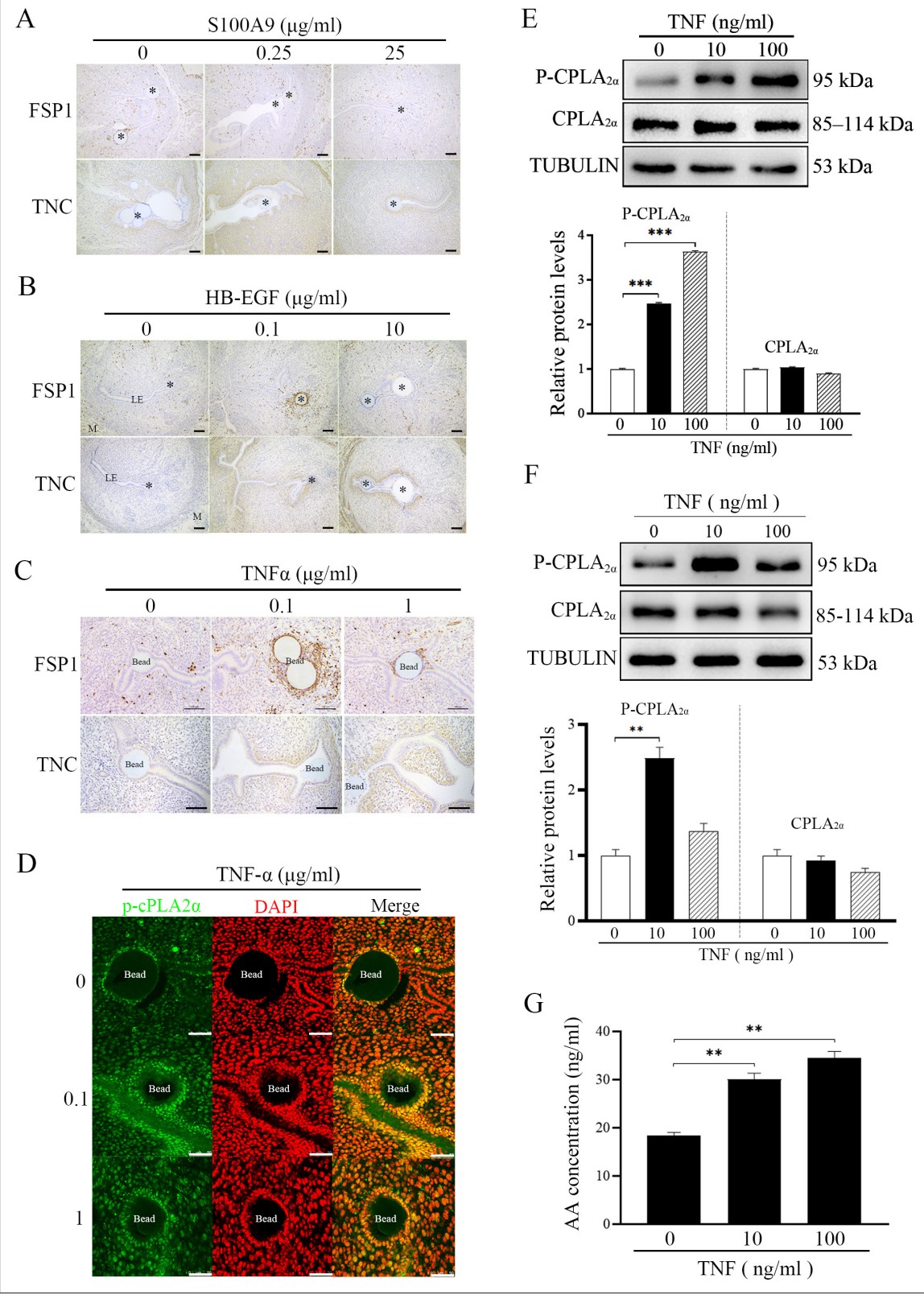

**Figure 6.** Effects of TNF on CPLA$_{2\alpha}$ phosphorylation and AA secretion. (**A**) Immunostaining of TNC and S100A4 in mouse uteri after S100A9-soaked blue beads were transferred into uterine lumen of PD4 mice for 24 hr (n=6 mice). (**B**) Immunostaining of TNC and S100A4 in mouse uteri after HB-EGF-soaked blue beads were transferred into uterine lumen of PD4 mice for 24 hr (n=6 mice). (**C**) Immunostaining of TNC and S100A4 in mouse uteri after TNF-soaked blue beads were transferred into uterine lumen of PD4 mice for 24 hr (n=6 mice).* Bead; LE, luminal epithelium; M, muscular layer; St,

*Figure 6 continued on next page*

*Figure 6 continued*

stroma. Scale bar = 100 µm. (**D**) P-CPLA$_{2\alpha}$ immunofluorescence in mouse uteri after 0.1 and 1 µg/ml TNF-soaked blue beads were transferred into PD4 uterine lumen (n=3 mice). Scale bar = 50 µm. (**E**) Western blot analysis of CPLA$_{2\alpha}$ and P-CPLA$_{2\alpha}$ protein levels after cultured epithelial cells were treated with TNF for 3 hr. (**F**) Western blot analysis of CPLA$_{2\alpha}$ and P-CPLA$_{2\alpha}$ protein levels after cultured epithelial organoids were treated with TNF for 3 hr. (**G**) ELISA analysis of AA concentration in the cultured medium after cultured epithelial organoids were treated with TNF for 6 hr. *, $p<0.05$; **, $p<0.01$; ***, $p<0.001$.

The online version of this article includes the following source data for figure 6:

**Source data 1.** Raw data of all western blots from *Figure 6*.

**Source data 2.** Complete and uncropped membranes of all western blots from *Figure 6*.

surgical specimens from 14 patients with uterine leiomyoma showed that TNC is significantly upregulated in patient samples (*Jamaluddin et al., 2019*). Fibrosis mostly occurs in pathological conditions. Intrauterine adhesions (IUA), also known as Asherman's syndrome, are caused by endometrial fibrosis as a result of injury to the uterus's basal lining, resulting in partial or total adhesions in the uterine cavity (*Healy et al., 2016*). IUA can interfere with the embryo's implantation and development, resulting in decreased or even full loss of intrauterine volume, female infertility, and recurrent miscarriages (*Leung et al., 2021*). The thin endometrial model exhibits a higher degree of fibrosis than normal controls, which is thought to be a crucial component in embryo implantation failure (*Zhang et al., 2022*). The key question is why fibrosis doesn't occur during normal early pregnancy? Both activins and inhibits are expressed in pregnant uterus (*Ni and Li, 2017*). In our study, *Fst* was significantly up-regulated under in vitro decidualization although ACTIVIN A was able to stimulate in vitro decidualization. FST is a secreted glycoprotein and can neutralize the profibrotic and proinflammatory actions of ACTIVINS. FST has a strong antifibrotic effect in various organs (*Aoki et al., 2005*; *Patella et al., 2006*). Furthermore, FST is shown to be critical for mouse decidualization (*Fullerton et al., 2017*). Additionally, AA was able to induce fibroblast activation and promote decidualization in our study. A recent study showed that 11,12-epoxyeicosatrienoic acid, a metabolite of AA can alleviate pulmonary fibrosis (*Kim et al., 2021*). It is possible that a physiological level of fibroblast activation is beneficial for decidualization and the long-lasting fibroblast activation could be balanced by certain molecules, like FST or AA metabolite.

## Conclusion

In this study, we identified that embryos-derived TNF is able to phosphorylate CPLA$_{2\alpha}$ for releasing AA from luminal epithelium. AA can physiologically induce fibroblast activation and promote decidualization via PGI$_2$-PPARδ-ACTIVIN A axis. This regulatory mechanism is also conserved in mice and humans. Overall, this study should shed a light on the novel mechanism underlying decidualization (*Figure 8J*).

## Materials and methods

**Key resources table**

| Reagent type (species) or resource | Designation | Source or reference | Identifiers | Additional information |
|---|---|---|---|---|
| Antibody | Anti-SPARC (rabbit monoclonal) | Cells Signaling Technology | Cat. #: 8725 S RRID:AB_2617210 | 1:200(IF) 1:1000(WB) |
| Antibody | Anti-Phospho-cPLA2 (rabbit polyclonal) | Cells Signaling Technology | Cat. #: 2831 S RRID:AB_2164445 | 1:100(IF) 1:1000(WB) |
| Antibody | Anti- Tenascin C (rabbit monoclonal) | Abcam | Cat. #: ab108930 RRID:AB_10865908 | 1:200(IHC) 1:1000(WB) |
| Antibody | Anti- S100A4 (rabbit monoclonal) | Cells Signaling Technology | Cat. #: 13018 S RRID:AB_10865908 | 1:200(IHC) 1:1000(WB) |
| Antibody | Anti- CPLA$_{2\alpha}$(rabbit polyclonal) | Santa Cruz Biotechnology | Cat. #: SC-438 RRID:AB_2164442 | 1:1000 |
| Antibody | Anti-α-SMA (rabbit monoclonal) | Cells Signaling Technology | Cat. #: 19,245T RRID:AB_476977 | 1:1000 |

*Continued on next page*

*Continued*

| Reagent type (species) or resource | Designation | Source or reference | Identifiers | Additional information |
|---|---|---|---|---|
| Antibody | Anti- COX-2 (rabbit monoclonal) | Cells Signaling Technology | Cat. #: 12,282T RRID:AB_1149420 | 1:1000 |
| Antibody | Anti- PGIS (rabbit polyclonal) | Cayman Chemical | Cat. #: 100023 RRID:AB_10816097 | 1:1000 |
| Antibody | Anti- PPAR delta (rabbit monoclonal) | Abcam | Cat. #: ab178866 RRID:AB_2165907 | 1:1000 |
| Antibody | Anti- BMP2 (rabbit polyclonal) | Abclonal | Cat. #: A0231 RRID:AB_2313822 | 1:1000 |
| Antibody | Anti- WNT4 (mouse monoclonal) | Santa Cruz Biotechnology | Cat. #: sc-376279 RRID:AB_787604 | 1:1000 |
| Antibody | Anti- E2F8 (rabbit polyclonal) | Abclonal | Cat. #: sc-376279 RRID:AB_10639701 | 1:1000 |
| Antibody | Anti- CYCLIN D3 (mouse monoclonal) | Cells Signaling Technology | Cat. #: 2936T RRID:AB_397675 | 1:1000 |
| Antibody | Anti-α- TUBULIN (mouse monoclonal) | Cells Signaling Technology | Cat. #: 2144 S RRID:AB_2716367 | 1:1000 |
| Antibody | Anti- GAPDH (mouse monoclonal) | Santa Cruz Biotechnology | Cat. #: sc-32233 RRID:AB_307275 | 1:1000 |
| Antibody | Goat anti-Rabbit IgG (H+L) Cross-Adsorbed Secondary Antibody, HRP (Goat polyclonal) | inventrogen | Cat. #: G21234 RRID:AB_2534776 | 1:5000 |
| Antibody | Alexa Fluor 488-AffiniPure Goat Anti-Rabbit IgG (H+L) (Goat polyclonal) | Jackson ImmunoResearch | Cat. #: 111-545-144 RRID:AB_2338046 | 1:200 |
| Chemical compound, drug | Recombinant Mouse TNF-alpha (aa 80–235) Protein | R&D systems | Cat. #: 410-MT-010 | 1–100 ng/ml(recombinant protein) |
| Chemical compound, drug | TNC | R&D systems | Cat. #: 410-MT-010 | 5–500 ng/ml(recombinant protein) |
| Chemical compound, drug | S100A4 | MedChemExpress | Cat. #: HY-P71084 | 50 and 500 ng/ml(recombinant protein) |
| Chemical compound, drug | SPARC | MedChemExpress | Cat. #: HY-P71086 | 1 and 10 µM (recombinant protein) |
| Chemical compound, drug | AA | Sigma | Cat. #: A3611 | 0.2 and 20 µM (recombinant protein) |
| Chemical compound, drug | ILOPROST | MedChemExpress | Cat. #: HY-A0096 | 0.1 and 10 µM (recombinant protein) |
| Chemical compound, drug | SELEXIPAG | MedChemExpress | Cat. #: HY-14870 | 0.1 and 10 µM (recombinant protein) |
| Chemical compound, drug | GW501516 | Cayman Chemical | Cat. #: 317318-70-0 | 0.1 and 10 µM (recombinant protein) |
| Chemical compound, drug | NS 398 | Selleck | Cat. #: S8433 | 20 and 40 µM(recombinant protein) |
| Chemical compound, drug | GSK0660 | Selleck | Cat. #: S5817 | 40 µMµM(recombinant protein) |
| Chemical compound, drug | ACTIVIN A | MedChemExpress | Cat. #: HY-P70311 | 1–100 ng/ml(recombinant protein) |
| Cell line (*Homo-sapiens*) | T HESCs | ATCC | CRL-4003 | |
| Cell line (*Homo-sapiens*) | Ishikawa | Chinese Academy of Science | CRL-2923 | |

*Continued on next page*

*Continued*

| Reagent type (species) or resource | Designation | Source or reference | Identifiers | Additional information |
|---|---|---|---|---|
| Commercial assay or kit | AA ELISA kit | Elabscience | Cat. #: E-EL-0051c | |
| Other | PI stain | Sigma | Cat. #: P4170 | 5 µg/ml |
| Other | DAPI stain | Sigma | Cat. #: D9542 | 20 µg/ml |
| Software, algorithm | SPSS | SPSS | RRID:SCR_002865 | |
| Software, algorithm | Image J | ImageJ (http://imagej.nih.gov/ij/) | RRID:SCR_003070 | |
| Software, algorithm | GraphPad Prism software | GraphPad Prism (https://graphpad.com) | RRID:SCR_015807 | Version 8.0.0 |

## Animals and treatments

Mature CD1 mice (Hsd: ICR mouse, 6 weeks of age) were purchased from Hunan Slaike Jingda Laboratory Animal Co., LTD and maintained in specific pathogen free (SPF) environment and controlled photoperiod (light for 12 hr and darkness for 12 hr). All animal protocols were approved by the Animal Care and Use Committee of South China Agricultural University (No. 2021f085). In order to induce pregnancy or pseudopregnancy, female mice aged 8–10 weeks were mated with male mice of reproductive age or vasectomized mice (vaginal plug day for day 1). To confirmed the pregnancy of female mice, embryos were flushed from fallopian tubes or uteri from days 1 to 4. On days 4 midnight (D4.5), 5 (D5) and 5 midnight (D5.5), implantation sites were identified by intravenous injection of 0.1 ml of 1% Chicago blue dye (Sigma-Aldrich, St. Louis, MO) dissolved in saline. Uteri were also collected from pseudopregnant mice on day 4 (PD4) , D4.5 , D5, and D5.5 .

On day 4 of pregnancy (0800–0900 h) (D4), pregnant mice were ovariectomized to induce delayed implantation. From days 5 to 7, progesterone was injected daily (1 mg/0.1 ml sesame oil/mice, Sigma-Aldrich) to maintain delayed implantation. Estradiol-17β (1 µg/ml sesame oil/mouse, MCE) (*Diao et al., 2008*) was subcutaneously injected on day 7 to activate embryo implantation. Delayed implantation was confirmed by flushing the blastocyst from the uterine horn. The implantation site of the activated uterus was determined by intravenous injection of Chicago blue dye.

## Transfer of TNF-soaked beads

Affi-Gel Blue Gel Beads (Bio-Rad # 1537302) with blastocyst size were incubated with TNF (0.1% BSA/PBS 410-MT-010, R&D systems, Minnesota, USA) 37 °C for 4 hr. After washed with embryo culture medium M2 (M7167, Sigma Aldrich, St. Louis, MO) (0.1% BSA) for three times, TNF-soaked beads (15 beads/horn) were transplanted into the uterine horns of PD4 mice (n=6 mice). Beads incubated in 0.1% BSA/PBS were used as control group (n=6 mice). Blue bands with beads were identified by injecting Chicago blue into the tail intravenous injection of Chicago blue to observe the implantation site 3 and 24 hr after transplantation, respectively.

## Immunofluorescence

Immunofluorescence was performed as described previously (*Li et al., 2020*; *Zheng et al., 2020*). Briefly, frozen sections (10 µm) were fixed in 4% paraformaldehyde (158127, Sigma Aldrich) for 10 min. Frozen or paraffin sections were blocked with 10% horse serum for 1 hr at 37 °C and incubated overnight with appropriate dilutions of primary antibodies at 4 °C. The primary antibodies used in this study included anti-SPARC (1:200, 8725 S, Cell Signaling Technology, Danvers, MA) and anti-P-CPLA$_{2\alpha}$ (1:100, 2831 S, Cell Signaling Technology). After washing in PBS, sections were incubated with secondary antibody (2.5 µg/ml, G21234, Invitrogen, Carlsbad, CA) for 40 min, counterstained with 4,6-diamidino-2-phenylindole dihydrochloride (20 µg/ml, DAPI, D9542, Sigma-Aldrich) or propidium iodide (5 µg/ml, PI, P4170, Sigma-Aldrich) and were mounted with ProLong Diamond Antifade Mountant (Thermo Fisher, Waltham, MA). The pictures were captured by laser scanning confocal microscopy (Leica, Germany). Each experiment was repeated at least three times.

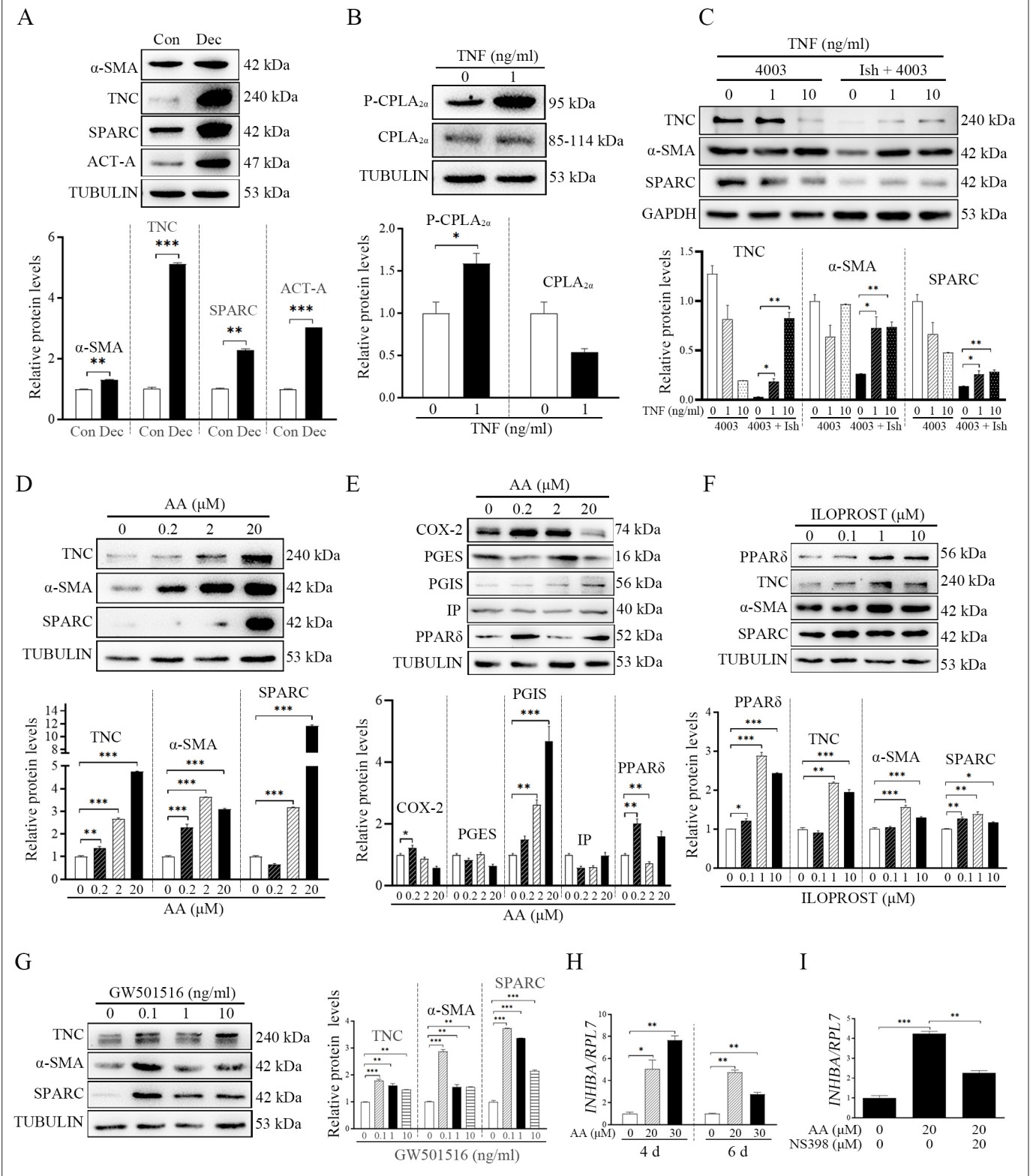

**Figure 7.** TNF regulation of fibroblast activation through AA-PGI-ACTIVIN A pathway in human endometrium. (**A**) Western blot analysis of α-SMA, TNC, SPARC, and ACTIVIN A protein levels after human stromal cells were induced for decidualization for 24 hr. ACT-A: ACTIVIN A. (**B**) Western blot analysis of CPLA₂α and P-CPLA₂α protein levels in Ishikawa cells after the co-culture of Ishikawa cells and 4003 cells were treated with TNF for 3 hr. (**C**) Western blot analysis of TNC, α-SMA, and SPARC protein levels in stromal 4003 cells after the co-culture of Ishikawa cells and 4003 cells were treated with TNF for 3 hr. Ish: Ishikawa. (**D**) Western blot analysis of TNC, α-SMA and SPARC protein levels after stromal 4003 cells were treated with AA for 6 hr.

*Figure 7 continued on next page*

Figure 7 continued

(**E**) Western blot analysis of COX-2, PGES, PGIS, and PPARδ protein levels after stromal 4003 cells were treated with AA for 3 hr. (**F**) Western blot analysis of PPARδ, TNC, α-SMA and SPARC protein levels after stromal 4003 cells were treated with ILOPROST for 12 hr. (**G**) Western blot analysis of TNC, α-SMA, and SPARC protein levels after stromal cells 4003 cells were treated with GW501516 for 6 hr. (**H**) QPCR analysis of *INHBA* mRNA levels after stromal 4003 cells were treated with AA. (**I**) QPCR analysis on effects of NS398 on AA stimulation of *INHBA* mRNA levels after stromal 4003 cells were treated with AA. *, $p < 0.05$; **, $p < 0.01$; ***, $p < 0.001$.

The online version of this article includes the following source data for figure 7:

**Source data 1.** Raw data of all western blots from *Figure 7*.

**Source data 2.** Complete and uncropped membranes of all western blots from *Figure 7*.

## Immunohistochemistry

Immunohistochemistry was performed as described previously (*Fu et al., 2019*). In short, paraffin sections (5 µm) were deparaffined, rehydrated, and antigen retrieved by boiling in 10 mM citrate buffer for 10 min. Endogenous horseradish peroxidase (HRP) activity was inhibited with 3% $H_2O_2$ solution in methanol. After washing for three times with PBS, the sections were incubated at 37°C for 1 hr in 10% horse serum for blocking, incubated overnight in each primary antibody at 4 °C. The primary antibodies used in this study included anti-TNC (1:200, ab108930, Abcam, Cambridge, UK) and anti-S100A4 (1:200, 13018 S, Cell Signaling Technology). After washing, the sections were incubated with biotinylated rabbit anti-goat IgG antibody (1:200, Zhongshan Golden Bridge, Beijing, China) and streptavidin-HRP complex (1:200, Zhongshan Golden Bridge). According to the manufacturer's protocol, the positive signals were visualized using DAB Horseradish Peroxidase Color Development Kit (Zhongshan Golden Bridge). The nuclei were counter-stained with hematoxylin. Each experiment was repeated at least three times.

## siRNA transfection

The siRNAs for mouse Activin A were designed and synthesized by Ribobio Co., Ltd. (Guangzhou, China). Following manufacturer's protocol, mouse endometrial stromal cells or human endometrial stromal cell 4003 (ATCC, CRL-4003) (American Type Culture Collection) were transfected with each *Inhba* siRNA using Lipofectamine 2000 Transfection Reagent (Invitrogen, Grand Island, NY) for 6 hr and 12 hr, respectively. The siRNA sequences were listed in *Table 1*. Each experiment was repeated at least three times.

## AA assay

Mouse uteri on D3, D4, and D4.5 (n=5 mice per day) were flushed with 0.4 ml of normal saline per mouse. The flushing solution from 5 mice was combined and dried in a vacuum freeze drier at –80 °C for 24 hr. The dried product was dissolved in 400 µl of the working buffer for AA measurement. After mouse epithelial organoids were treated with TNF (10 and 100 ng/ml) for 6 hr (n=3), the cultured medium was collected for AA measurement. AA amount in the uterine fluid or cultured medium was measured with AA ELISA kit according to the manufacturer's instruction (Elabscience, E-EL-0051c, Wuhan, China). This kit's sensitivity is greater than 0.94 ng/ml. In brief, 50 µl of each sample was incubated at 37 °C for 45 min with 50 µl of biotinylated antibody working solution, 100 µL of HRP enzyme conjugate working solution for 30 min, and 90 µl of substrate solution for 15 min before being stopped with 50 µl of substrate solution. The solution was immediately read at 450 nm with a Biotek microplate reader (ELX808). Absorbance values for AA standards were calculated in the same way. The AA standard curve was used to calculate the concentrations of AA.

## Isolation and treatment of mouse uterine luminal epithelial cells

Uterine luminal epithelial cells were isolated as previously described (*Nallasamy et al., 2012*). The uteri from the estrous mice or PD4 were cut longitudinally, washed in HBSS, incubated in 0.2% (W/V) trypsin (0458, Amresco, Cleveland, USA) and 6 mg/ml dispase (Roche Applied Science,4942078001, Basel, Switzerland) in 4.3 ml HBSS for 1.5 hr at 4°C, 30 min at room temperature, and 10 min at 37°C. After rinsing in HBSS, the epithelial cells were precipitated in 5% BSA in HBSS for 7 min. After the collected epithelial cells were cultured in DMEM/F12 (D2906, Sigma-Aldrich) with 10% heat-inactivated fetal bovine serum (FBS) in a culture plate for 30 min, the unattached epithelial cells were

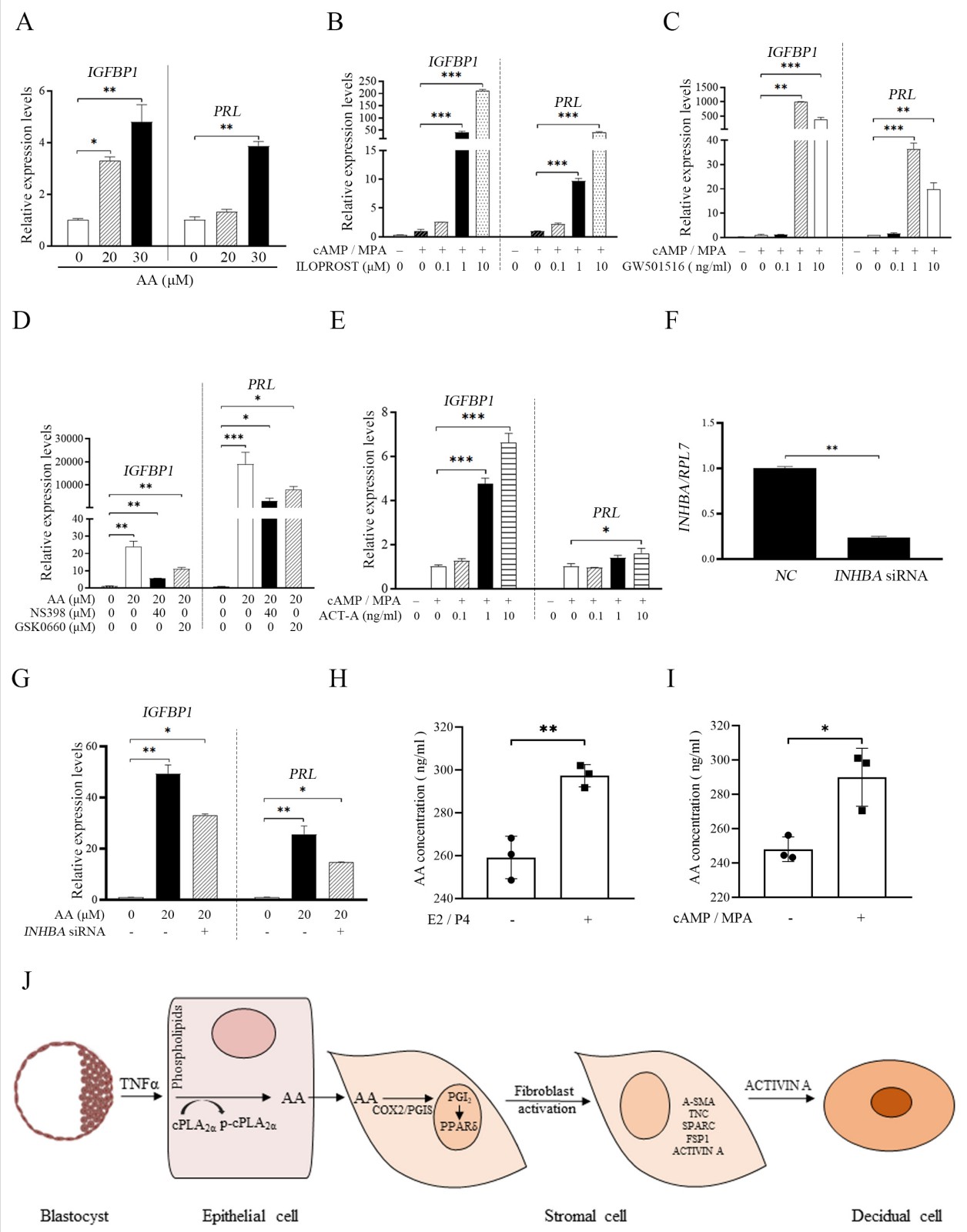

**Figure 8.** AA-PGI2-PPARδ-ACTIVIN A regulation on human decidualization. (**A**) QPCR analysis of *IGFBP1* and *PRL* mRNA levels after stromal 4003 cells were treated with AA. (**B**) QPCR analysis of *IGFBP1* and *PRL* mRNA levels after stromal 4003 cells were treated with ILOPROST under in vitro decidualization for 4 days. (**C**) QPCR analysis of *IGFBP1* and *PRL* mRNA levels after stromal 4003 cells were treated with GW501516 under in vitro decidualization for 4 days. (**D**) QPCR analysis on effects of NS398 or GSK0660 on AA induction of *IGFBP1* and *PRL* mRNA levels after stromal 4003 cells

*Figure 8 continued on next page*

*Figure 8 continued*

were treated with AA for 4 days. (**E**) QPCR analysis of *IGFBP1* and *PRL* mRNA levels after stromal 4003 cells were treated with ACTIVIN A for 2 days under in vitro decidualization. ACT-A: ACTIVIN A. (**F**) QPCR analysis on the interference efficiency of *INHBA*. (**G**) QPCR analysis on effects of *INHBA* siRNAs on AA induction of *IGFBP1* and *PRL* mRNA levels in stromal 4003 cells for 4 days. (**H**) AA concentration in the cultured medium after mouse stromal cells were induced for in vitro decidualization with estrogen (E2) and progesterone (P4) for 24 h. (**I**) AA concentration in the cultured medium after human endometrial stromal 4003 cells were induced for in vitro decidualization with MPA and cAMP for 4 days. (**J**) The model of fibroblast activation action in the blastocyst-endometrial interaction. *, p<0.05; **, p<0.01; ***, p<0.001.

transferred into new culture plates precoated with ECM (1:100, E0282, Sigma-Aldrich) and cultured at 37 °C for 1 hr. Luminal epithelial cells were treated with TNF (10 and 100 ng/ml, 410-MT-010, R&D systems) in DMEM/F12 with 2% charcoal-treated FBS (cFBS, Biological Industries, Cromwell, CT).

## Culture and treatment of mouse uterine luminal epithelial organoids

Uterine luminal epithelial organoids were isolated and cultured as previously described (*Turco et al., 2017*; *Boretto et al., 2017*; *Luddi et al., 2020*). Mouse uteri on PD4 was dissected longitudinally, rinsed three times in HBSS, and incubated in 3.5 ml of HBSS containing 100 mg/ml Trypsin and 60 mg/ml dispase at 4 °C for 1 hr, room temperature for 1 hr, and 37 °C for 10 min. The epithelium was washed out in HBSS after digested, collected, centrifuged, and precipitated. After removing the supernatant, epithelial cells were suspended in DMEM/F12 medium and adjusted to the density of $1.5 \times 10^7$ cells /ml. The epithelial cell suspension was mixed with ice-precooled ECM (1:3, 356231, BD biocoat, Becton-Dickinson, MA). The epithelial cells suspended in ECM were seeded onto the preheated 24-well plate and cultured in CO2 incubator at 37 °C with Organoid medium, including 1% ITS-G (PB180429, Procell, Wuhan, China), 2 mM L-glutamine (49419, Sigma-Aldrich), 1 mM nicotinamide (49419, Sigma-Aldrich), 2% B27 (17504–044, Gibco, Grand Island, NY), 1% N2 (17502–048, Gibco), 50 ng/ml EGF (HY-P7067, MedChemExpress, NJ, USA),100 ng/ml FGF-basic (HY-P7066, MedChemExpress), 100 ng/ml Noggin (HY-P70785, MedChemExpress), 200 ng/ml WNT-3A (315–20, Peprotech, Rocky Hill, USA), 200 ng/ml R-Spondin-1 (HY-P76012, MedChemExpress), and 0.5 µM A83-01 (HY-10432, MedChemExpress) in DMEM/F12. The organoids with lumen were formed after 6–7 days culture and treated with TNF (10 and 100 ng/ml) for 3 or 6 hr.

## Isolation and treatment of mouse endometrial stromal cells

Mouse endometrial stromal cells were isolated as previously described (*Zheng et al., 2020*). Briefly, mouse uteri on PD4 were cut longitudinally, washed in HBSS, and incubated with 1% (W/V) trypsin and 6 mg/ml dispase in 3.5 mL HBSS for 1 hr at 4 °C, for 1 hr at room temperature and for 10 min at 37 °C. The uterine tissues were washed with Hanks' balanced salt solution, incubated in 6 ml of HBSS containing 0.15 mg/ml Collagenase I (Invitrogen, 17100–017) at 37 °C for 35 min. Primary endometrial stromal cells were cultured with DMEM/F12 containing 10% FBS.

Mouse endometrial stromal cells were induced for in vitro decidualization as previously described (*Liang et al., 2014*). Primary endometrial stromal cells were treated with 10 nM of Estradiol-17 β and 1 µM of P4 in DMEM/F12 containing 2% charcoal-treated FBS (cFBS, Biological Industries) to induce decidualization in vitro for 72 hr. Stromal cells were treated with TNC (5, 50 and 500 ng/ml, 3358-TC-050, R&D systems), S100A4 (50 and 500 ng/ml, HY-P71084, MedChemExpress), SPARC (1 and 10 µM, HY-P71086, MedChemExpress), AA (0.2, 2 and 20 µM, A3611, Sigma-Aldrich), PGI analogue ILOPROST (0.1, 1 and 10 µM, HY-A0096, MedChemExpress), SELEXIPAG (0.1, 1 and 10 µM, HY-14870, MedChemExpress), PPARδ agonist GW501516 (0.1, 1 and 10 µg/ml, 317318-70-0, Cayman Chemical, Ann Arbor, MI), COX-2 antagonist NS 398 (20 and 40 µM, S8433, Selleck, Shanghai, China), PPARδ antagonist GSK0660 (40 µM, 1014691-61-2, Selleck), and ACTIVIN A (1, 10 and 100 ng/ml, HY-P70311, MedChemExpress) in DMEM/F12 containing 2% cFBS, respectively.

## Culture and treatment of human cell lines

Human endometrial adenocarcinoma Ishikawa cells (Chinese Academy of Science, Shanghai, China) and human endometrial stromal 4003 cells (CRL-4003, American Type Culture Collection) were cultured in DMEM/F12 with 10% FBS, and supplemented with 100 units/ml penicillin and 0.1 mg/ml streptomycin (PB180429, Procell) at 37 °C, 5% CO2, respectively. TNF (1 and 10 ng/ml) was used to treat Ishikawa cells. For further experiments, cultured 4003 cells were treated with AA (0.2, 2 and

**Table 1.** Primers and siRNA sequences used in this study.

| Gene | Species | Sequence (5'–3') | Application | Accession Number | Product size |
|------|---------|------------------|-------------|------------------|--------------|
| Rpl7 | Mouse | GCAGATGTACCGCACTGAGATTC ACCTTTGGGCTTACTCCATTGATA | RT-QPCR | NM_011291.5 | 129 bp |
| Prl8a2 | Mouse | AGCCAGAAATCACTGCCACT TGATCCATGCACCCATAAAA | RT-QPCR | NM_010088 | 119 bp |
| Inaba | Mouse | CCAGTCTAGTGCTTCTGGGC GATGAGGGTGGTCTTCGGAC | RT-QPCR | NM_008380.2 | 156 bp |
| RPL7 | Human | CTGCTGTGCCAGAAACCCTT TCTTGCCATCCTCGCCAT | RT-QPCR | NM_000971 | 194 bp |
| IGFBP1 | Human | CCAAACTGCAACAAGAATG GTAGACGCACCAGCAGAG | RT-QPCR | NM_001013029 | 87 bp |
| PRL | Human | AAGCTGTAGAGATTGAGGAGCAAA TCAGGATGAACCTGGCTGACTA | RT-QPCR | NM_000948 | 76 bp |
| INHBA | Human | TCATCACGTTTGCCGAGTCA TGTTGGCCTTGGGGACTTTT | RT-QPCR | NM_002192 | 129 bp |
| NC | - | CTCCGAACGTGTCACGT | siRNA | | |
| Activin a | Mouse | GAACAGTCACATAGACCTT | siRNA | | |
| ACTIVIN A | Human | CCAUGUCCAUGUUGUACUATT | siRNA | | |

20 µM), PGI analogue ILOPROST (0.1, 1 and 10 µM), PPARδ agonist GW501516 (0.1, 1 and 10 µg/ml) and ACTIVIN A (1, 10 and 100 ng/ml). The STR profiling of both Ishikawa cells and 4003 cells was used to confirm the identity of these cell lines. Both Ishikawa cells and 4003 cells were tested for mycoplasma-free.

## Co-culture of epithelial cells and stromal cells

Human Ishikawa cells were cultured in 30 mm culture plates. Human 4003 cells were cultured on glass slipper. After reaching 70–80% confluence, human 4003 cells cultured on glass slippers were transferred into Ishikawa culture plates and supported with 4 prefixed plastic pillars. Human 4003 cells on glass slipper were co-cultured with the underlying Ishikawa cells in one culture plate. After the co-cultured 4003 cells and Ishikawa cells were treated with TNF, 4003 cells and Ishikawa cells were further analyzed, respectively.

## Western blot

Western blot was performed as previously described (*Li et al., 2020*). The primary antibodies used in this study included $CPLA_{2\alpha}$ (1:1000, SC-438, Santa Cruz Biotechnology, Dallas, TX), phosphorylated $CPLA_{2\alpha}$ (1:1000), TNC (1:1000), SPARC (1:1000), α-SMA (1:1000, 19,245T, Cell Signaling Technology), COX-2 (1:1000, 12,282T, Cell Signaling Technology), PGIS (1:1000, 100023, Cayman Chemical), PPARδ (1:1000, ab178866, Abcam), BMP2 (1:1000, A0231, Abclonal, Wuhan, China), WNT4 (1:1000, sc-376279, Santa Cruz Biotechnology), E2F8 (1:1000, A1135, Abclonal), CYCLIN D3 (1:1000, 2936T, Cell Signaling Technology), TUBULIN (1:1000, 2144 S, Cell Signaling Technology), GAPDH (1:1000, sc-32233, Santa Cruz Biotechnology). After the membranes were incubated with an HRP-conjugated secondary antibody (1:5000, Invitrogen) for 1 hr, the signals were detected with an ECL Chemiluminescent Kit (Millipore, USA). Each experiment was repeated at least three times.

## Real-time RT-PCR

The total RNA was isolated using the Trizol Reagent Kit (9109, Takara, Japan), digested with RQ1 deoxyribonuclease I (Promega, Fitchburg, WI), and reverse-transcribed into cDNA with the Prime Script Reverse Transcriptase Reagent Kit (Takara, Japan). For real-time PCR, the cDNA was amplified using a SYBR Premix Ex Taq Kit (TaKaRa) on the CFX96 Touch Real-Time System (Bio-Rad). For real-time PCR System (Bio-Rad). Data were analyzed using the 2-△△Ct method and normalized to *Rpl7*

(mouse) or *RPL7* (human) level. The corresponding primer sequences of each gene were provided in *Table 1*. Each experiment was repeated at least three times.

## Statistical analysis

The data were analyzed by GraphPad Prism8.0 Student's T test was used to compare differences between two groups. The comparison among multiple groups was performed by ANOVA test. All the experiments were repeated independently at least three times. In the mouse study, each group had at least three mice. Data were presented as mean ± standard deviation (SD). A value of $p < 0.05$ was considered significant.

## Acknowledgements

Funding This study was supported by National Natural Science Foundation of China (32171114 and 31871511) and National Key Research and Development Program of China (2018YFC1004400).

## Additional information

### Funding

| Funder | Grant reference number | Author |
| --- | --- | --- |
| National Natural Science Foundation of China | 32171114 | Zeng-Ming Yang |
| National Key Research and Development Program of China | 2018YFC1004400 | Zeng-Ming Yang |
| National Natural Science Foundation of China | 31871511 | Zeng-Ming Yang |

The funders had no role in study design, data collection and interpretation, or the decision to submit the work for publication.

### Author contributions

Si-Ting Chen, Conceptualization, Data curation, Formal analysis, Investigation, Methodology, Writing – original draft; Wen-Wen Shi, Yu-Qian Lin, Ying Wang, Data curation, Investigation; Zhen-Shan Yang, Data curation, Formal analysis, Investigation; Meng-Yuan Li, Yue Li, Data curation, Validation, Investigation; Ai-Xia Liu, Yali Hu, Conceptualization, Data curation, Writing – review and editing; Zeng-Ming Yang, Conceptualization, Supervision, Funding acquisition, Writing – original draft, Project administration, Writing – review and editing

### Author ORCIDs

Si-Ting Chen ⓘ http://orcid.org/0000-0001-5127-0236
Zeng-Ming Yang ⓘ https://orcid.org/0000-0001-6775-0082

### Ethics

All animal protocols were approved by the Animal Care and Use Committee of South China Agricultural University (No. 2021f085).

### Decision letter and Author response

Decision letter https://doi.org/10.7554/eLife.82970.sa1
Author response https://doi.org/10.7554/eLife.82970.sa2

## Additional files

### Supplementary files
MDAR checklist

Source data 1. Uncropped western blot images used in this study.

## Data availability

All data generated or analysed during this study are included in the manuscript. Source data files are provided for Figures 1-7.

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
