## [Editor Report]

The authors provide novel evidence for a connection between fibroblast activation and eutherian stromal decidualization. This important work substantially advances our understanding of decidua biology and its contribution to pregnancy. The authors are using solid evidence to support the findings.

---

## [Decision Letter]

**Decision letter after peer review:**

Thank you for submitting your article "Fibroblast activation during decidualization: Embryo-derived TNFα induction of PGI_2_-PPARδ-ACTIVIN A pathway through luminal epithelium" for consideration by *eLife*. Your article has been reviewed by 3 peer reviewers, and the evaluation has been overseen by a Reviewing Editor and Ricardo Azziz as the Senior Editor. The following individual involved in the review of your submission has agreed to reveal their identity: Günter P. Wagner (Reviewer #2).

Essential revisions:

1. The abstract need to be restructured, specifically the section objective does not list objectives as such, but is rather a series of unconnected sentences that have more to do with an introduction. The objectives paragraph should include the objective. The authors finish the section with the hypothesis, but the objective is not directly mentioned.

2. The authors must rewrite the introduction. The current introduction is full of different ideas not interconnected, which makes it difficult to follow the background for understanding methods and results. Perhaps, the authors can use the same style that in the Discussion section which is clearer and better structured. I suggest the next flow:

Start with endometrial decidualization, which is the major topic of the paper, and finish this part by mentioning that although the underlying mechanisms are not fully described there is a similarity with fibroblast activation. Then, the authors can explain the background of fibroblast activation, including the markers that are assessed in the study. The question in the discussion "why fibrosis doesn't occur during normal early pregnancy?" can be included to remark on a striking point of their hypothesis. Finally, once the authors set up what is known and unknown, they can finish with the last paragraph which is appropriate if the information exposed before is improved.

3. Cheng et al., have used several techniques to test their hypothesis. However, there are some aspects of material and methods that should be addressed to promote the understanding and reproducibility of results. First of all, in every section, it must be clarified the type of sample on which the technique is applied. For example, in line 165, immunofluorescence in which frozen sections are applied. Additionally, in immunohistochemistry and immunostaining, it must appear the antibody dilution in each molecule tested as well as secondary antibodies. In this section, it is very important to detail the experimental steps in order to be able to repeat the test if necessary. In the section "Isolation and treatment of mouse endometrial cells" there is a lack of information regarding the cell types used and the concentrations of molecules tested. Although they appear in the Results section, they should also be included in this section specifying that different concentrations were used and even naming a reference article where these concentrations are used, if available.

4. Technical and biological replicates must appear in Materials and methods and results to check the robustness and reproducibility of the assays performed and the results presented. Especially, they should include the sample size of human donors.

5. Concentration of the reagents used must be included in the methodology to ensure the reproducibility of the assays.

6. Authors should always refer to treatment days in the same way. For example, when they refer to day 4 or day 5 of pregnancy (line 292) the fists time should appear day 4 (D4) and the rest of the time should always be referred to as D4 as well as in the case where they refer to hours, in some cases they use the terminology morning, midnight and other times the specific hour, authors may use always the same term.

7. In figure 1 (line 299) the molecule POSTN does not appear in the results.

8. In line 315 authors mentioned SPARC overexpression, why it does not appear in the Materials and methods section? and why they use a plasmid in this case with this molecule? It seems that is out of context, authors should clarify this decision.

9. Why do authors not test the classical Prl8a2 and Alpl as classical decidualization markers in mice decidualization? The base of the paper is culture cell decidualization, thus the authors should robustly demonstrate their cultures are decidualized. The Prl8a2 just appears in figure 4.

10. In the presentation of the results the authors refer to changes in some genes or proteins as "significant" where they presumably mean a p-value of <0.05. However, it is more important to report the estimated effect size (fold change, difference, or similar) than just "significance." Limiting the reporting of experimental results to p-values is a major problem in the scientific literature and needs to be changed to include reference to effect sizes.

11. The authors should use siRNA knockdown experiments to prove Activin A role in the process. The authors should reevaluate the epithelial coculture experiments and Ishikawa experiments. Maybe including no epithelial cells in the coculture experiments as a control will validate their use of these cells Finally remove the Trisomy 16 experiments or develop them further.

The authors should use siRNA knockdown experiments to prove Activin A role in the process. The authors should reevaluate the epithelial coculture experiments and Ishikawa experiments. Maybe including no epithelial cells in the coculture experiments as a control will validate their use of these cells Finally remove the Trisomy 16 experiments or develop them further.

*Reviewer #1 (Recommendations for the authors):*

The experimental design used is confused with the mice model and human primary/cell lines. The Materials and methods section is not well structured, and the level of detail does not allow the replication of the experiments carried out (dose, time, dilutions, statistics performed) to check the robustness and reproducibility of the assays performed and the results presented. There is a lack of consistency between the time of exposure in different experiments (3h, 6h, 12h, 24h), as well as, decidualization stimulus (E2+P4 or cAMP+MPA) used and time of exposure (24h, 2d, 4d, 5d…) that is a basic requirement for the comparison between experiments. Finally, the experiments need to show more detail in the graphs bar showing the values for each sample and indicating if it is a sample or experimental replicate, controls/normalization values used in each experiment, quantification of IHC and IF experiments, and all the western blot used for calculated protein quantification in the supplemental material (from the different experiments performed). Finally, the experiment using decidua samples with fetal 16 trisomy is very interconnected and there is a lack of clinical detail of these samples and which objective is being followed to include them.

1. In my opinion, the abstract need to be restructured, specifically the section objective does not list objectives as such but is rather a series of unconnected sentences that have more to do with an introduction. The objectives paragraph should include the objective. The authors finish the section with the hypothesis, but the objective is not directly mentioned.

2. The authors must rewrite the introduction. The current introduction is full of different ideas not interconnected, which makes it difficult to follow the background for understanding methods and results. Perhaps, the authors can use the same style that in the Discussion section which is clearer and better structured. I suggest the next flow:

Start with endometrial decidualization, which is the major topic of the paper, and finish this part by mentioning that although the underlying mechanisms are not fully described there is a similarity with fibroblast activation. Then, the authors can explain the background of fibroblast activation, including the markers that are assessed in the study. The question in the discussion "why fibrosis doesn't occur during normal early pregnancy?" can be included to remark on a striking point of their hypothesis. Finally, once the authors set up what is known and unknown, they can finish with the last paragraph which is appropriate if the information exposed before is improved.

3. Cheng et al., have used several techniques to test their hypothesis. However, there are some aspects of material and methods that should be addressed to promote the understanding and reproducibility of results. First of all, in every section, it must be clarified the type of sample on which the technique is applied. For example, in line 165, immunofluorescence in which frozen sections are applied. Additionally, in immunohistochemistry and immunostaining, it must appear the antibody dilution in each molecule tested as well as secondary antibodies. In this section, it is very important to detail the experimental steps in order to be able to repeat the test if necessary. In the section "Isolation and treatment of mouse endometrial cells" there is a lack of information regarding the cell types used and the concentrations of molecules tested. Although they appear in the Results section, they should also be included in this section specifying that different concentrations were used and even naming a reference article where these concentrations are used, if available.

4. Technical and biological replicates must appear in Materials and methods and results to check the robustness and reproducibility of the assays performed and the results presented. Especially, they should include the sample size of human donors.

5. Concentration of the reagents used must be included in the methodology to ensure the reproducibility of the assays.

6. Authors should always refer to treatment days in the same way. For example, when they refer to day 4 or day 5 of pregnancy (line 292) the fists time should appear day 4 (D4) and the rest of the time should always be referred to as D4 as well as in the case where they refer to hours, in some cases they use the terminology morning, midnight and other times the specific hour, authors may use always the same term.

7. In figure 1 (line 299) the molecule POSTN does not appear in the results.

8. In line 315 authors mentioned SPARC overexpression, why it does not appear in the Materials and methods section, and why do they use a plasmid in this case with this molecule? It seems that is out of context, authors should clarify this decision.

9. Why do authors not test the classical Prl8a2 and Alpl as classical decidualization markers in mice decidualization? The base of the paper is culture cell decidualization, thus the authors should robustly demonstrate their cultures are decidualized. The Prl8a2 just appears in figure 4.

*Reviewer #2 (Recommendations for the authors):*

In the presentation of the results, the authors refer to changes in some genes or proteins as "significant" where they presumably mean a p-value of <0.05. However, it is more important to report the estimated effect size (fold change, difference, or similar) than just "significance." Limiting the reporting of experimental results to p-values is a major problem in the scientific literature and needs to be changed to include reference to effect sizes.

*Reviewer #3 (Recommendations for the authors):*

The authors should use siRNA knockdown experiments to prove Activin A role in the process. The authors should reevaluate the epithelial coculture experiments and Ishikawa experiments. Maybe including no epithelial cells in the coculture experiments as a control will validate their use of these cells Finally remove the Trisomy 16 experiments or develop them further.

[Editors’ note: further revisions were suggested prior to acceptance, as described below.]

Thank you for resubmitting your work entitled "Embryo-derive TNF promotes decidualization via fibroblast activation" for further consideration by *eLife*. Your revised article has been evaluated by Diane Harper (Senior Editor) and a Reviewing Editor.

The manuscript has been improved but there are some remaining issues that need to be addressed, as outlined below:

Lack of effect of PGE2 on FB activation: the effect of PGE2 depends on the PTGER that is expressed in those cells. It has been shown that decidualization is specifically dependent on PTGER2/EP2. In human ESF EP2 expression is induced by progesterone, so the question is whether the PGE2 experiment was done in the presence of progesterone/MPA.

*Reviewer #2 (Recommendations for the authors):*

This revision is addressing most of my concerns and my assessment is that this is an important paper. The main finding is that endometrial stromal cells undergo a two step transformation during decidualization, first a step that leads to a state homologous to fibroblast activation and in a second step the activated fibroblast state converts into the decidual cell type. This model has been suggested before, but this paper is adding a large amount of experimental evidence and maps a particular causal pathway involved in this process.

While this work is well done and important, it does raise obvious questions with respect to previous results that at least should be discussed, even if only so briefly. In their paper the authors say that the signal for decidualization is arachidonic acid (AA) produced in the luminal epithelial cells and show that AA in vitro is sufficient to induce at least FB activation. In the classical in vitro model human ESF are decidualized by MPA and cAMP, and there is no AA present as far as one would expect. How do the authors explain that discrepancy? Is the AA pathway parallel to the MPA cAMP, where the latter stands for some kind of signal acting through a GPCR?

Line 51: the phrase "local cAMP" is still not adequate, since it is specifically the production of cAMP in the ESF activating PKA. "Local" is too unspecific.

Line 84: "attachment to the luminal epithelium" rather than "attachment onto the uterus". The latter could mean to the outside of the uterus.

Line 102: what does CD1 is referring to here? CD1 is a protein, and PPARd is not CD1.

Line 109: Embryo-derived.

Line 118: FSP1 is not a clear designation, since there are at least three genes with this alias (S100A4, ATL1, AIFM2). Please use unique signifiers. On line 412, the authors reveal that they are talking about S110A4, but that is way too late in the manuscript. And confusing aliases should be avoided generally.Figure 3 A, and line 181: again, these experiments with PGE2 seem to be done in the absence of P4 and are thus irrelevant since PGE2 receptor expression is controlled by progesterone.

The textual description of the results in Figures 3 D to G hides a much more complex picture of concentration dependent up and downregulation.

Line 208: write out “Arachidonic Acid” at least in the title of a section and only then use an acronym.

Line 427: the term “COX2 induced PGI2” is inaccurate. COX2 does not “induce” PGI2, but produces the precursor for ALL prostaglandins. In this context “PGI2” alone is sufficient.

Line 458: better “A proper coordination between embryo attachment and decidualization…". Relate two processes (attachment and decidualization) rather than a thing (the embryo) and a process (decidualization).

Lines 459 to 468: all of that is correct but seems tangential to the findings reported in this paper.

Line 47“: "Ashman's" or "Asherman's".

---

## [Author Response]

Essential revisions:1. The abstract need to be restructured, specifically the section objective does not list objectives as such, but is rather a series of unconnected sentences that have more to do with an introduction. The objectives paragraph should include the objective. The authors finish the section with the hypothesis, but the objective is not directly mentioned.

Yes, the abstract was revised according to your requirement.

2. The authors must rewrite the introduction. The current introduction is full of different ideas not interconnected, which makes it difficult to follow the background for understanding methods and results. Perhaps, the authors can use the same style that in the Discussion section which is clearer and better structured. I suggest the next flow:Start with endometrial decidualization, which is the major topic of the paper, and finish this part by mentioning that although the underlying mechanisms are not fully described there is a similarity with fibroblast activation. Then, the authors can explain the background of fibroblast activation, including the markers that are assessed in the study. The question in the discussion "why fibrosis doesn't occur during normal early pregnancy?" can be included to remark on a striking point of their hypothesis. Finally, once the authors set up what is known and unknown, they can finish with the last paragraph which is appropriate if the information exposed before is improved.

Yes, the introduction was revised according to your requirement.

3. Cheng et al., have used several techniques to test their hypothesis. However, there are some aspects of material and methods that should be addressed to promote the understanding and reproducibility of results. First of all, in every section, it must be clarified the type of sample on which the technique is applied. For example, in line 165, immunofluorescence in which frozen sections are applied. Additionally, in immunohistochemistry and immunostaining, it must appear the antibody dilution in each molecule tested as well as secondary antibodies. In this section, it is very important to detail the experimental steps in order to be able to repeat the test if necessary. In the section "Isolation and treatment of mouse endometrial cells" there is a lack of information regarding the cell types used and the concentrations of molecules tested. Although they appear in the Results section, they should also be included in this section specifying that different concentrations were used and even naming a reference article where these concentrations are used, if available.

Yes, the detailed information was provided in our revised version.

4. Technical and biological replicates must appear in Materials and methods and results to check the robustness and reproducibility of the assays performed and the results presented. Especially, they should include the sample size of human donors.

Yes, the information on replicates was provided in our revised version.

5. Concentration of the reagents used must be included in the methodology to ensure the reproducibility of the assays.

Yes, the concentration of the reagents used was provided.

6. Authors should always refer to treatment days in the same way. For example, when they refer to day 4 or day 5 of pregnancy (line 292) the fists time should appear day 4 (D4) and the rest of the time should always be referred to as D4 as well as in the case where they refer to hours, in some cases they use the terminology morning, midnight and other times the specific hour, authors may use always the same term.

Yes, the terms were revised according to your requirements.

7. In figure 1 (line 299) the molecule POSTN does not appear in the results.

Sorry for this mistake. POSTN was deleted.

8. In line 315 authors mentioned SPARC overexpression, why it does not appear in the Materials and methods section? and why they use a plasmid in this case with this molecule? It seems that is out of context, authors should clarify this decision.

Yes, we agree with you. We newly bought recombinant SPARC protein and redid this treatment. The results from SPARC overexpression was replaced by recombinant SPARC in Figure 2E.

9. Why do authors not test the classical Prl8a2 and Alpl as classical decidualization markers in mice decidualization? The base of the paper is culture cell decidualization, thus the authors should robustly demonstrate their cultures are decidualized. The Prl8a2 just appears in figure 4.

Yes, we agree with you. Data from Prl8a2 were provided in Figures 2B, 2D, 2F, 2I, 3F, and 3H.

10. In the presentation of the results the authors refer to changes in some genes or proteins as "significant" where they presumably mean a p-value of <0.05. However, it is more important to report the estimated effect size (fold change, difference, or similar) than just "significance." Limiting the reporting of experimental results to p-values is a major problem in the scientific literature and needs to be changed to include reference to effect sizes.

Yes, we agree with you. The fold changes were provided in our revised version.

11. The authors should use siRNA knockdown experiments to prove Activin A role in the process. The authors should reevaluate the epithelial coculture experiments and Ishikawa experiments. Maybe including no epithelial cells in the coculture experiments as a control will validate their use of these cells Finally remove the Trisomy 16 experiments or develop them further.

Yes, siRNA knockdown on Activin A was performed and data was provided in Figure 8F and 8G.

Reviewer #1 (Recommendations for the authors):1. In my opinion, the abstract need to be restructured, specifically the section objective does not list objectives as such but is rather a series of unconnected sentences that have more to do with an introduction. The objectives paragraph should include the objective. The authors finish the section with the hypothesis, but the objective is not directly mentioned.

Yes, the abstract was revised according to your requirement.

2. The authors must rewrite the introduction. The current introduction is full of different ideas not interconnected, which makes it difficult to follow the background for understanding methods and results. Perhaps, the authors can use the same style that in the Discussion section which is clearer and better structured. I suggest the next flow:Start with endometrial decidualization, which is the major topic of the paper, and finish this part by mentioning that although the underlying mechanisms are not fully described there is a similarity with fibroblast activation. Then, the authors can explain the background of fibroblast activation, including the markers that are assessed in the study. The question in the discussion "why fibrosis doesn't occur during normal early pregnancy?" can be included to remark on a striking point of their hypothesis. Finally, once the authors set up what is known and unknown, they can finish with the last paragraph which is appropriate if the information exposed before is improved.

Yes, the introduction was revised according to your requirement.

3. Cheng et al., have used several techniques to test their hypothesis. However, there are some aspects of material and methods that should be addressed to promote the understanding and reproducibility of results. First of all, in every section, it must be clarified the type of sample on which the technique is applied. For example, in line 165, immunofluorescence in which frozen sections are applied. Additionally, in immunohistochemistry and immunostaining, it must appear the antibody dilution in each molecule tested as well as secondary antibodies. In this section, it is very important to detail the experimental steps in order to be able to repeat the test if necessary. In the section "Isolation and treatment of mouse endometrial cells" there is a lack of information regarding the cell types used and the concentrations of molecules tested. Although they appear in the Results section, they should also be included in this section specifying that different concentrations were used and even naming a reference article where these concentrations are used, if available.

Yes, the detailed information was provided in our revised version.

4. Technical and biological replicates must appear in Materials and methods and results to check the robustness and reproducibility of the assays performed and the results presented. Especially, they should include the sample size of human donors.

Yes, the information on replicates was provided in our revised version.

5. Concentration of the reagents used must be included in the methodology to ensure the reproducibility of the assays.

Yes, the concentration of the reagents used was provided.

6. Authors should always refer to treatment days in the same way. For example, when they refer to day 4 or day 5 of pregnancy (line 292) the fists time should appear day 4 (D4) and the rest of the time should always be referred to as D4 as well as in the case where they refer to hours, in some cases they use the terminology morning, midnight and other times the specific hour, authors may use always the same term.

Yes, the concentration of the reagents used was provided.

7. In figure 1 (line 299) the molecule POSTN does not appear in the results.

Sorry for this mistake. POSTN was deleted.

8. In line 315 authors mentioned SPARC overexpression, why it does not appear in the Materials and methods section, and why do they use a plasmid in this case with this molecule? It seems that is out of context, authors should clarify this decision.

Yes, we agree with you. We newly bought recombinant SPARC protein and redid this treatment. The results from SPARC overexpression was replaced by recombinant SPARC in Figure 2E.

9. Why do authors not test the classical Prl8a2 and Alpl as classical decidualization markers in mice decidualization? The base of the paper is culture cell decidualization, thus the authors should robustly demonstrate their cultures are decidualized. The Prl8a2 just appears in figure 4.

Yes, we agree with you. Data from Prl8a2 were provided in Figures 2B, 2D, 2F, 2I, 3F, and 3H.

Reviewer #2 (Recommendations for the authors):In the presentation of the results, the authors refer to changes in some genes or proteins as "significant" where they presumably mean a p-value of <0.05. However, it is more important to report the estimated effect size (fold change, difference, or similar) than just "significance." Limiting the reporting of experimental results to p-values is a major problem in the scientific literature and needs to be changed to include reference to effect sizes.

Yes, we agree with you. The fold changes were provided in our revised version.

Reviewer #3 (Recommendations for the authors):The authors should use siRNA knockdown experiments to prove Activin A role in the process. The authors should reevaluate the epithelial coculture experiments and Ishikawa experiments. Maybe including no epithelial cells in the coculture experiments as a control will validate their use of these cells Finally remove the Trisomy 16 experiments or develop them further.

Yes, siRNA knockdown on Activin A was performed and data was provided in Figure 8F and 8G.

[Editors’ note: what follows is the authors’ response to the second round of review.]The manuscript has been improved but there are some remaining issues that need to be addressed, as outlined below:Lack of effect of PGE2 on FB activation: the effect of PGE2 depends on the PTGER that is expressed in those cells. It has been shown that decidualization is specifically dependent on PTGER2/EP2. In human ESF EP2 expression is induced by progesterone, so the question is whether the PGE2 experiment was done in the presence of progesterone/MPA.

Yes, we agree with you. When mouse stromal cell were treated with PGE2 under in vitro decidualization (estrogen and progesterone), there was an increase of TNC and SPARC, but PGE2 had no obvious effect on αSMA (A). This result was provided in Figure 3A. When human stromal cells were treated with PGE2 under in vitro decidualization (MPA and dbcAMP), PGE2 had no obvious effects on the protein levels of αSMA, TNC and SPARC (B).

Based on these results, it seems that PGI2 action on fibroblast activation is stronger than PGE2. The effects of PGE2 on fibroblast activation should be slightly different from decidualization induction. As you mentioned, PGE2 should be important for human in vitro decidualization.

Reviewer #2 (Recommendations for the authors):This revision is addressing most of my concerns and my assessment is that this is an important paper. The main finding is that endometrial stromal cells undergo a two step transformation during decidualization, first a step that leads to a state homologous to fibroblast activation and in a second step the activated fibroblast state converts into the decidual cell type. This model has been suggested before, but this paper is adding a large amount of experimental evidence and maps a particular causal pathway involved in this process.While this work is well done and important, it does raise obvious questions with respect to previous results that at least should be discussed, even if only so briefly. In their paper the authors say that the signal for decidualization is arachidonic acid (AA) produced in the luminal epithelial cells and show that AA in vitro is sufficient to induce at least FB activation. In the classical in vitro model human ESF are decidualized by MPA and cAMP, and there is no AA present as far as one would expect. How do the authors explain that discrepancy? Is the AA pathway parallel to the MPA cAMP, where the latter stands for some kind of signal acting through a GPCR?

Yes, we agree with you. Although our study indicated that AA-PGI2 pathway induce fibroblast activation and decidualization, previous studies also showed that PGI2 is able to activate cAMP signaling pathway (Beck et al., 2014; Fetalvero et al., 2008). It is possible that PGI2 may also induce fibroblast activation and decidualization through cAMP pathway. Additionally, our study also confirmed AA secretion during either mouse or human in vitro decidualization (Figure 8H for mouse and I for human in vitro decidualization, respectively), which may have positive feedback during in vitro decidualization. According to your requirements, we provided this information in Discussion section of our revised version.

Beck, F., Geiger, J., Gambaryan, S., Veit, J., Vaudel, M., Nollau, P., et al., 2014. Timeresolved characterization of cAMP/PKA-dependent signaling reveals that platelet inhibition is a concerted process involving multiple signaling pathways. Blood 123(5):e1-e10.

Fetalvero, K.M., Zhang, P., Shyu, M., Young, B.T., Hwa, J., Young, R.C., et al., 2008. Prostacyclin primes pregnant human myometrium for an enhanced contractile response in parturition. J Clin Invest 118(12):3966-3979.

Line 51: the phrase "local cAMP" is still not adequate, since it is specifically the production of cAMP in the ESF activating PKA. "Local" is too unspecific.

Yes, we agree with you. "Local" have been changed to " local cyclic adenosine monophosphate (cAMP) from endometrial stromal cells ".

Line 84: "attachment to the luminal epithelium" rather than "attachment onto the uterus". The latter could mean to the outside of the uterus.

Yes, we agree with you. "onto the uterus" have been changed to "to the luminal epithelium".

Line 102: what does CD1 is referring to here? CD1 is a protein, and PPARd is not CD1.

Sorry for this mistake. CD1 refers to the mouse strain, which we have deleted.

Line 109: Embryo-derived.

Sorry for this mistake. " Embryos-derived " have been changed to " Embryo-derived ".

Line 118: FSP1 is not a clear designation, since there are at least three genes with this alias (S100A4, ATL1, AIFM2). Please use unique signifiers. On line 412, the authors reveal that they are talking about S110A4, but that is way too late in the manuscript. And confusing aliases should be avoided generally.

Sorry for this mistake. FSP1 was replaced by S100A4. S100A4 (also called FSP1) was also provided when S100A4 first appeared.

Figure 3 A, and line 181: again these experiments with PGE2 seem to be done in the absence of P4 and are thus irrelevant since PGE2 receptor expression is controlled by progesterone.

Yes, we have added the effect of PGE2 on fibroblast activation in the presence of progesterone in Figure 3A.

The textual description of the results in Figures 3 D to G hides a much more complex picture of concentration dependent up and downregulation.

Sorry for this mistake. According to your requirements, we provided a detailed description on this result.

Line 208: write out “Arachidonic Acid” at least in the title of a section and only then use an acronym.

Sorry for this mistake. We have revised accordingly.

Line 427: the term “COX2 induced PGI2” is inaccurate. COX2 does not “induce” PGI2, but produces the precursor for ALL prostaglandins. In this context “PGI2” alone is sufficient.

Yes, we agree with you. " COX2 induced PGI2" have been changed to " PGI2".

Line 458: better “A proper coordination between embryo attachment and decidualization…”. Relate two processes (attachment and decidualization) rather than a thing (the embryo) and a process (decidualization).

Yes, we agree with you. We have revised accordingly.

Lines 459 to 468: all of that is correct but seems tangential to the findings reported in this paper.

Yes, we agree with you. We have removed this part of the content.

Line 47“: "Ashman's" or "Asherman's".

Sorry for this mistake. " Ashman's " have been changed to " Asherman's ".